# Domain adapted brain network fusion captures variance related to pubertal brain development and mental health

Dominik Kraft [1] ✉, Dag Alnæs[2] & Tobias Kaufmann [1,2,3] ✉

Puberty demarks a period of profound brain dynamics that orchestrates changes to a multitude of neuroimaging-derived phenotypes. This complexity poses a dimensionality problem when attempting to chart an individual's brain development over time. Here, we illustrate that shifts in subject similarity of brain imaging data relate to pubertal maturation in the longitudinal ABCD study. Given that puberty depicts a critical window for emerging mental health issues, we additionally show that our model is capable of capturing variance in the adolescent brain related to psychopathology in a population-based and a clinical cohort. These results suggest that low-dimensional reference spaces based on subject similarities render useful to chart variance in brain development in youths.

Recent availability of big data in the neurosciences and sparking technical advances have opened doors toward a system level understanding of high-dimensional, multimodal data, integrating information from genetic, behavioral, and neuroimaging sources, amongst others[1]. Such deep phenotyping avenues are holding great promise to unravel the complexity and heterogeneity of mental disorders, where a multitude of factors have been identified as contributors to the risk architectures and clinical phenotypes[2–4]. Multimodal big data, however comes with the curse of dimensionality[5] or hurdles regarding how to efficiently and effectively integrate different information sources in biologically meaningful manners[6,7].

Previous research has approached the task of data integration from various angles, from data concatenation to sophisticated modeling[7] such as similarity network fusion (SNF[8]). First application attempts of SNF to common brain disorders have illustrated its potential for deriving insights from heterogeneous populations such as those with psychiatric (e.g.[9]) or neurological (e.g.[6]) disorders. SNF is an unsupervised technique that integrates unique and complementary information from different data sources, thus placing individuals in a comprehensive and biologically informed feature space, which is defined by the similarity between subjects across all data modalities. To achieve this, SNF exploits the covariance between data modalities. Subsequent dimensionality reduction methods such as diffusion map

embedding[10] may reveal dominant axes of inter-subject similarity on which subjects can be localized by a single score.

Similar attempts of charting an individual's position on a data continuum have recently shown success in psychiatry, where mapping dimensions of psychopathology can yield advantages over categorical systems, e.g.[11,12].

A key challenge in dimensions that are based on inter-subject similarity is that newly added samples can inevitably result in a change to the overall similarity structure. Consequently, the score that localizes an individual on the dimension is not stable as would be desirable in biomarker utilities, thus marking a disadvantage compared to other data-derived markers such as polygenic risk scores or measures of brain structure. To overcome this, we here propose a machine learning (ML) framework that learns the mapping from raw structural MRI features to the low-dimensional brain embedding score and through supervised domain adaptation allows to transfer this mapping into new datasets without the need to recalculate the fused network. Figure 1A describes the framework schematically. Our approach comes with advantages over modeling fused networks independently for individual datasets and timepoints: First, our ML model establishes a subject similarity reference space in an independent training sample, allowing for robust predictions at an individual

[1]Department of Psychiatry and Psychotherapy, Tübingen Center for Mental Health, University of Tübingen, Tübingen, Germany. [2]Norwegian Centre for Mental Disorders Research, University of Oslo and Oslo University Hospital, Oslo, Norway. [3]German Center for Mental Health (DZPG), partner site Tübingen, Tübingen, Germany. ✉e-mail: Dominik.Kraft@med.uni-tuebingen.de; tobias.kaufmann@medisin.uio.no

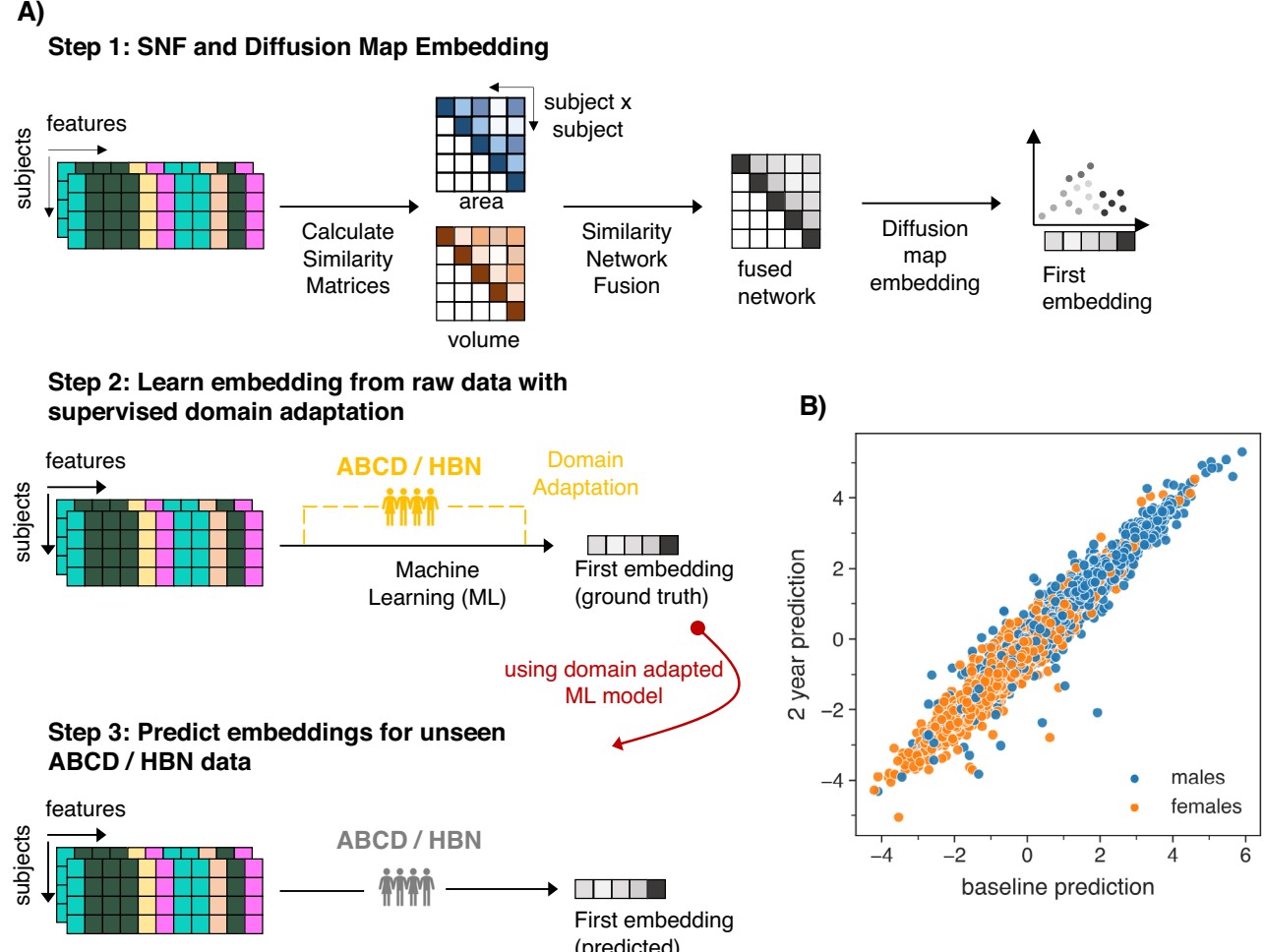

**Fig. 1 | Inferring a reference space using domain adaptation on brain network embeddings. A** Schematic workflow of the prediction framework. Step 1: Similarity network fusion is followed by diffusion map embedding to extract individual subject scores of the first brain embedding. Step 2: A machine learning model is trained to learn the mapping between raw features and the first brain embedding score. Using domain adaptation, held-out subsets of the target data (yellow) from ABCD or HBN are added to the training, respectively. Step 3: Such domain adaptation enhances out of sample prediction for unseen (gray) data in both datasets. **B** Predicted brain embeddings for the ABCD baseline (*x*-axis) and 2-years follow-up (*y*-axis) data reveal a sex gradient. PNC Philadelphia Neurodevelopmental Cohort, ABCD Adolescent Brain Cognitive Development, HBN Healthy Brain Network.

subject's level in unseen data. Second, domain adaptation offers flexibility to adapt the model to other datasets that have unique characteristics, such as repeated measures in a longitudinal design or heterogeneity that is commonly found in patient samples. To this end, we trained our model in the Philadelphia Neurodevelopmental Cohort (PNC)[13] and withheld data from the target datasets that was used for domain adaptation. We validated our approach in an unseen longitudinal sample from the Adolescent Brain Cognitive Development (ABCD) Study[14] and on a clinical population of subjects from the Healthy Brain Network (HBN)[15] sample. Both datasets allow to investigate unique processes shaping the human brain in development, specifically pubertal maturation, and emerging psychopathology.

Puberty depicts a phase of biological and psychological changes potentially mediated by neurodevelopment beyond the effect of age[16–18]. Variables assessing pubertal status can thus be more sensitive measures than age for studying brain maturation in youth (e.g.,[19]). Previous work revealed global reductions in cortical gray matter volumes and thickness with advanced pubertal maturation, with evidence from both, cross-sectional and longitudinal data. These effects appear to be distributed across the whole cortex rather than being circumscribed to a specific set of regions (see ref. 17 for a review). However, as different studies use different approaches to account for age and sex, inconsistencies exist in terms of effect sizes and effect directions, including those of opposing effect directions in males and females[20,21]. These conflicting observations might arise from certain methodological choices but also from individual variability in pubertal timing and progression through maturational stages. While all adolescents undergo the same pubertal stages, there is quite some variability regarding pubertal onset and tempo of changes, which has been linked to mental health conditions[22–24]. In females, earlier pubertal timing appears to be associated to worse mental health conditions (e.g.,[22,25]), while for boys both very early and very late onset has been linked to worse psychological outcome (e.g.,[26,27]).

Given the close interplay between pubertal maturation, brain development, and its link to emerging psychopathology, we aimed at investigating the sensitivity of brain embeddings toward these two entities. We show that our model captures variance of pubertal brain development and allows to capture biological variance related to emerging psychopathology, suggesting its utility in investigating within-person changes in youths.

## Results

### Model performance

We applied SNF with subsequent diffusion map embedding to data from $N = 1594$ individuals spanning a wide developmental age range (8–21 years, PNC[13]).

Akin to other dimensionality reduction approaches, the first brain embedding captures most variance and was therefore used to build the reference space, referred to as brain embedding. We trained a machine learning model with an instance-based domain adaptation procedure (Transfer AdaBoost for Regression)[28] in a combined sample comprising the PNC sample and held-out data from ABCD or HBN to learn the mapping between raw MRI features—specifically cortical area and volume—and the brain embedding. This yielded a domain adapted reference model that could be applied to independent data in the ABCD and HBN samples. For the ABCD test sample, we applied the model on baseline and 2-years follow-up data, yielding two predictions per participant. For the cross-sectional HBN sample, the model yielded one prediction per participant. Model performance was calculated by comparing the predicted brain embeddings in the ABCD and HBN dataset to the 'ground truth' brain embeddings after performing SNF and diffusion map embedding on the respective test datasets. Our model achieved high performance in unseen data, both for the ABCD and HBN sample (Table 1). Brain maps illustrating the associations between brain embeddings and raw features showed similar patterns in both samples (Supplementary Fig. 1). Model performance was better in the ABCD sample, which might be driven by the fact, that the sample for domain adaptation in the ABCD dataset was approximately 5× larger than the one used for HBN, allowing for a more efficient shift towards the target distribution. Furthermore, the HBN set comprised data from patients, thus the lower accuracy may to some degree also reflect pathological variance. Moreover, within the ABCD sample,

baseline performance was slightly better compared to the follow-up data, since the data used for domain adaptation was also from the baseline study visit. Given successful performance of the model, we proceeded to validating the biological signal in the predictions.

### Biological validation of the model

We validated the biological utility of the predictions in capturing developmental brain dynamics by targeting puberty and mental health as two phenotypes that are closely related to each other. They both lay off their dynamics during adolescence and therefore are also intertwined with (developmental) brain trajectories[24,29]. We hypothesized that these phenotypes should be related to our brain embedding score. To assess the models' ability to capture variance cross-sectionally, we first calculated puberty associations for both timepoints and their respective brain embeddings in the ABCD sample, accounting the statistical model for age and scan site. We observed associations between the average puberty score measured with the Pubertal Development Scale (PDS)[30] and the predicted brain embedding at both timepoints for the caregiver reports (baseline$_{female}$: $b = -0.34$, $p = 6.88 \times 10^{-16}$, $\eta^2 = 0.02$, $N = 3344$; baseline$_{male}$: $b = -0.36$, $p = 5.53 \times 10^{-10}$, $\eta^2 = 0.01$, $N = 3920$; follow-up$_{female}$: $b = -0.27$, $p = 1.94 \times 10^{-15}$, $\eta^2 = 0.03$, $N = 3316$; follow-up$_{male}$: $b = -0.17$, $p = 1.39 \times 10^{-15}$, $\eta^2 = .008$, $N = 3910$). In youth reports we observed similar effects although some did not survive Bonferroni correction (baseline$_{female}$: $b = -0.17$, $p = 0.005$, $\eta^2 = .006$, $N = 1479$; baseline$_{male}$: $b = -0.06$, $p = 0.34$, $\eta^2 = 0.0005$, $N = 2264$; follow-up$_{female}$: $b = -0.20$, $p = 2.66 \times 10^{-9}$, $\eta^2 = 0.02$, $N = 3271$; follow-up$_{male}$: $b = -0.14$, $p = 0.0003$, $\eta^2 = 0.006$, $N = 4056$; see Fig. 2). Aiming at replicating these puberty associations in the clinical HBN sample, we performed two additional analyses in which we subsampled the HBN sample to the age of the ABCD baseline and the ABCD follow-up data. Calculating the same cross-sectional puberty models in these HBN subsets did not yield statistically significant results (Supplementary Data 1).

Beyond the cross-sectional associations, the framework allows to apply the model to longitudinal data of the same subjects and investigate change scores between timepoints, as the predicted brain embedding is modeled with respect to the reference and thus remains stable compared to fused networks derived from individual timepoints, which might introduce additional variance when computing the difference score. Accordingly, we argue that the difference between two predicted brain embeddings (Δ brain embedding) is capable of tracing brain trajectories and thus may serve as a marker for

**Table 1 | Model performance for unseen data in the ABCD and HBN sample**

| | RMSE | MAE | $R^2$ | r |
|---|---|---|---|---|
| ABCD$_{baseline}$ | 0.95 | 0.85 | 0.79 | 0.94 |
| ABCD$_{follow-up}$ | 1.02 | 0.91 | 0.78 | 0.94 |
| HBN | 1.50 | 0.95 | 0.65 | 0.92 |

*RMSE* root-mean-squared error, *MAE* mean absolute error, $R^2$ coefficient of determination, *r* Pearson correlation coefficient.

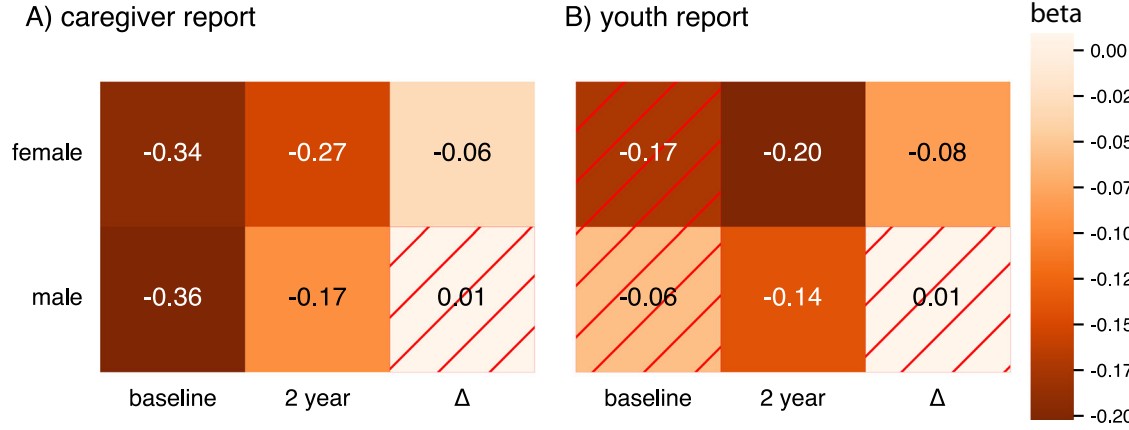

**Fig. 2 | Associations between brain embeddings and puberty, both in cross-sectional and longitudinal data.** First two columns in (**A**) and (**B**) refer to associations between predicted brain embeddings and the respective pubertal score (PDS mean) per timepoint. Δ refers to the association between the Δ brain embedding and the Δ PDS mean score. Annotations refer to effect sizes (betas) and hashed cells indicate non-significant results.

brain dynamics. Hence, we were particularly interested whether the Δ brain embedding captures biologically meaningful pubertal variance and is thus sensitive to biologically relevant processes shaping the human brain. Consequently, we repeated the linear models with Δ brain embedding as dependent and the Δ PDS scores (i.e., the puberty difference between baseline and 2-years follow-up) as independent variable (caregiver report: female mean = 0.77, male mean = 0.38, youth report: female mean = 0.70, male mean = 0.21). For age adjustment of this longitudinal analysis, we included the age difference between baseline and the 2 years follow-up (Δ age) as a covariate.

Whereas cross-sectional effect sizes were comparable between sexes across both timepoints, change association appeared to be more pronounced in females. For females we observed significant associations between Δ PDS and Δ brain embedding for both caregiver ($b = -0.06$, $p = 2.37 \times 10^{-10}$, $\eta^2 = 0.02$, $N = 3135$) and youth report ($b = -0.08$, $p = 3.79 \times 10^{-11}$, $\eta^2 = 0.04$, $N = 1375$) whereas for males, associations did not pass adjustment for multiple comparison (Bonferroni-adjusted $\alpha = 0.05/12 = 0.004$; caregiver: $b = 0.01$, $p = 0.19$, $\eta^2 = 0.0002$, $N = 3700$; youth: $b = 0.01$, $p = 0.36$, $\eta^2 = 0.0003$, $N = 2204$; see Fig. 2). These effects for females were even more pronounced after controlling for baseline puberty status (caregiver: $b = -0.08$, $p = 7.88 \times 10^{-20}$, $\eta^2 = 0.02$, youth: $b = -0.11$, $p = 2.38 \times 10^{-18}$, $\eta^2 = 0.04$). It is worth noting that whereas age explained some variance in the brain embedding scores, significant pubertal effects were always larger than the respective age effects (Supplementary Data 2), supporting that the brain embedding captures variance relevant to pubertal development beyond age related brain changes. After accounting for Body Mass

Index (BMI), socioeconomic status (SES), and race/ethnicity in the puberty association models, associations between Δ PDS and Δ brain embedding remained significant whereas cross-sectional associations did not (Supplementary Fig. 2 for methodological details and Supplementary Data 2 for exact model outcomes), further supporting sensitivity of the approach to longitudinal contexts.

Related, we observed that Δ brain embeddings are distributed quite equally across males and females for early pubertal stages, whereas from the 'midpubertal' period onwards distributions start to diverge with respect to earlier developmental stages but also with respect to between group differences (Fig. 3). Interestingly, deviations between sexes get even more pronounced with females' menarche, that marks the onset of the late pubertal state.

Puberty and adolescence depict a time of cascading changes ranging from biological, emotional to social domains and this phase of transition also constitutes a sensitive and critical period for emerging psychopathology and mental disorders[24,31,32]. Assuming that mental disorders emerge as deviations from a brain 'norm'[33] we argue that our approach of modeling the low-dimensional representation anchored to a population sample may allow to exploit the resulting reference space (i.e., the brain embedding) in a normative fashion. To validate this, we tested in a sample of patients drawn from the HBN[15] cohort for associations between the predicted brain embedding score and mental health. We calculated a proxy measure for psychopathology severity, that is the sum of all diagnoses per subject. Participants had between 1 and 10 diagnoses (mean$_{male}$ = 2.71, std$_{male}$ = 1.62, mean$_{female}$ = 2.71, std$_{female}$ = 1.55). Using this proxy measure as independent variable

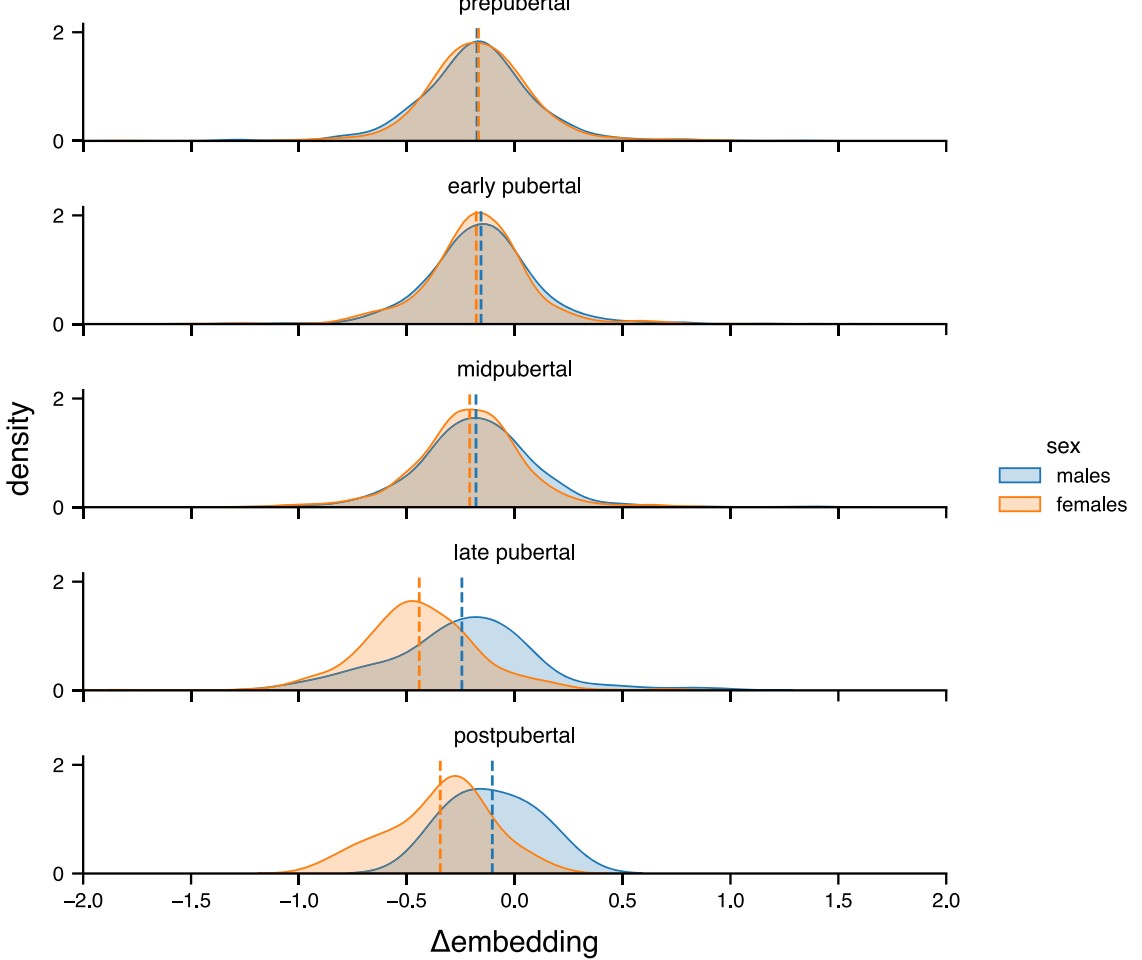

**Fig. 3 | Distribution of Δ brain embedding in the ABCD sample stratified for sex and pubertal categories at one-year follow-up.** Pubertal categories are based on youth report, but caregiver-based categories follow the same pattern (Supplementary Fig. 3). Vertical dashed lines indicate the mean Δ brain embedding per group.

together with age and site as covariates, we did not observe a significant effect of psychopathology on the brain embedding for males ($b = 0.04$, $p = 0.05$, $\eta^2 = 0.001$, $N = 1487$), but for females the effect survived multiple comparison correction ($b = 0.07$, $p = 0.007$, $\eta^2 = 0.007$, $N = 784$; Bonferroni-adjusted $\alpha = 0.05/2 = 0.025$). The identified association remained significant when covarying for PDS (females: $b = 0.07$, $p = 0.01$, $\eta^2 = 0.01$, $N = 589$), yet PDS itself was not significant in this cross-sectional sample, nor were interaction terms between puberty and psychopathology (see Supplementary Data 1 for all effects). Replacing the sum of diagnosis with a dimensional measure of psychopathology, i.e., the CBCL total score (Child-Behavior Checklist[34]) we replicated the effects from the previous analysis. Specifically, we did not observe a significant effect of psychopathology on the brain embedding for males ($b = 0.002$, $p = 0.06$, $\eta^2 = 0.004$, $N = 1269$), however, for females the effect reached statistical significance ($b = 0.004$, $p = 0.008$, $\eta^2 = 0.01$, $N = 635$). The identified association for psychopathology operationalized via the CBCL total score in females remained significant when controlling for puberty (PDS): $b = 0.005$, $p = 0.002$, $\eta^2 = 0.02$, $N = 471$. Since dimensional measures of psychopathology allow to test these associations also in a non-clinical sample, we aimed at validating these findings in the cross-sectional and longitudinal data from the ABCD sample. Here, we observed significant effects (all $p < 0.005$) between the CBCL score and the brain embedding, for all cross-sectional models (males and female) and the longitudinal model in males (Supplementary Fig. 4 and Supplementary Data 2 for details and exact effect sizes).

## Discussion

The present work illustrates a proof of concept for an approach that allows to map high high-dimensional brain imaging data into a low-dimensional brain embedding score which can be then transferred to new datasets by means of domain adaptation and machine learning. By doing so, our framework builds upon similarity network fusion[8] integrating information from different data sources, but does not suffer under the instability of similarity measures and thus can be translated to datasets with unique features such as longitudinal study designs or clinical cohorts without the need to recalculate a fused network in the new sample.

To validate our framework and to test its applicability to other datasets, we trained our model in a sample of subjects spanning a wide age range from the PNC cohort[13] with simultaneous supervised domain adaptation and tested it on two independent validation samples, that is longitudinal data from the ABCD Study[14] and a clinical population of subjects from the HBN sample[15]. Domain adaptation in both datasets was enhanced with independent data that was later not used in the prediction process, such as data from ABCD subjects for whom only baseline data was available and participants in the HBN cohort without clinical diagnoses. Model performance was high for unseen test data in both datasets, confirming the model's ability to generalize to other cohorts. Our approach thus proved useful in two unseen datasets that both displayed unique sample characteristics. We hypothesize that the good model performance also relies on choosing the PNC sample as a source task which stores a rich repertoire of (dis)similarities between participants, from which the domain adaptation procedure for the two new datasets could have benefitted. However, we consider it important to further investigate the frameworks' boundaries in terms of sample characteristics of the source and target datasets, that is, under which condition the model performance diminishes.

Beyond model building, we aimed at investigating whether brain embedding scores are sensitive to capture biologically meaningful variance in processes shaping the brain and thus may represent a useful imaging phenotype for (developmental) brain dynamics. Related to work suggesting a close link between pubertal dynamics and neurodevelopment[19,29,35], we observed significant cross-sectional associations between the predicted brain embedding scores and

puberty measures for all models at all timepoints except for baseline data based on youth reports, which might have been biased by the difficulty to rate one´s own pubertal maturation at these early ages. Such bias appears to be particularly true for males at baseline (see Supplementary Fig. 5). In addition, we observed higher correlations between caregiver and youth reports for the 2-year follow-up, suggesting an overall better alignment between reports, potentially minimizing biases. Of note, all analyses were performed stratified for sex, because the brain embeddings span a sex-gradient (Fig. 1B), and pubertal timing and trajectories are known to vary between females and males[36]. Moreover, for models in which we observed significant puberty effects, these effects were always larger than the respective age effects, supporting its sensitivity to puberty specific dynamics beyond age.

After adding BMI, SES, and race/ethnicity[37,38] as covariates into our model, cross-sectional effects diminished. This aligns with reports suggesting a close link between those factors and pubertal timing and duration (e.g.,[38,39]). Given the high inter-correlation between the studied variables, it may be difficult to disentangle variance to distinct components. Therefore, we argue, that longitudinal analyses may help to resolve the ambiguity of the cross-sectional analyses. Since the ABCD study offers an unprecedented resource for granular investigations of child and adolescent brain and pubertal maturation, we leveraged the longitudinal data of the ABCD cohort and investigated whether the Δ brain embedding, that is the difference between the two predicted brain embedding scores for baseline and the 2-years follow-up data, can serve as an additional marker for brain trajectories. Pubertal associations with the Δ brain embeddings were significant for females, but not for males, which appeared to align with the pubertal maturation in females in the studied time period. The same pattern was observable when controlling for BMI, SES, and race/ethnicity, supporting that the reported cross-sectional puberty effects do not simply represent differences in these confounding factors either.

Moreover, it appears that the Δ brain embeddings for both sexes follow a comparable distribution in early pubertal stages, whereas from females´ menarche onwards, both patterns start to deviate from each other. Given the narrow age range in the ABCD study, pubertal categories may serve as a proxy for pubertal timing with females often undergoing earlier puberty. Greater pubertal stage for a given age has been related to more mature, i.e., thinner cortices (e.g.[22]), which might be a putative explanation for the divergence in the Δ brain embedding. Upcoming releases of the ABCD data may help to further investigate the Δ brain embedding and its ability to capture subtle biological processes like pubertal maturation. With additional longitudinal data one would also expect to have access to more datapoints that represent male participants in later pubertal stages potentially allowing to better disentangle the putative brain trajectories encoded by the Δ brain embedding. While studies on normative brain development generally report overarching brain trajectories across different brain measures[40], recent work by Bottenhorn and colleagues[41] highlight a high degree of intra- and interindividual variability in brain maturation across imaging measures. Identifying and understanding these sources of variance depict an important step towards population-level neuroscience, which however may complicate downstream analyses because of the heterogeneity across regions and imaging measures. Because of its sensitivity to pubertal processes shaping the human brain, we suggest that our approach may help to unify those different sources of variance into a condensed score that does not only serve as a dimension reduction technique but places individuals in a biologically meaningful feature space.

Since puberty is a critical time window for emerging mental disorders[24,31,32], we aimed at additionally exploiting the models predictions as a 'normative' score and tested its association to psychopathology in the HBN sample. In females only, we observed small yet

significant effects of psychopathology severity on the brain embedding score.

By accounting these analyses for age, we ruled out that these associations simply mimic a larger number of diagnoses with increasing age. Since descriptive statistics of the number of diagnoses were almost identical across sexes (Supplementary Fig. 6), we deem it unlikely that such subtle variances might have driven differences in association strength. A more likely explanation might be the diagnoses themselves, as we observed diagnosis distributions matching the known patterns of more male-prevalent (e.g., ADHD) vs. more female-prevalent (e.g., mood or anxiety) disorders. Thus, it may be possible that the derived brain embeddings are more sensitive to female-prevalent disorders. Even though puberty associations did not reach statistical significance in the HBN sample, they pointed towards higher effects in females. As many female-typical mental disorders emerge during puberty[42] our brain embedding may also have a higher sensitivity towards those disorders. However, since we were not able to test this directly because of sample size restrictions, this line of reasoning should be considered as hypothesis generating and needs to be investigated in future research. When extending the model with pubertal variables, we did not observe additional effects, which may be related to prevalence of emerging psychopathology in the HBN sample, which may interfere with puberty and may thus explain, why we were not able to replicate the cross-sectional puberty associations in the HBN sample. Furthermore, subsampling the HBN sample to the respective age ranges of the ABCD visits decreased the available sample size in a way that has not sufficient statistical power to detect small to moderate effects.

We acknowledge that the sum of diagnoses in the HBN sample rather depicts a coarse measure of psychopathology, however, by expanding our work with a dimensional measure (the CBCL total psychopathology score), we did observe similar associations. Furthermore, both measures were moderately correlated ($r \sim 0.3$) supporting our initial approach to operationalize sum of diagnosis as a measure of psychopathology severity. Future research may leverage more fine-grained quantities, such as hierarchical representations of psychopathology (HiTOP[43]) or different syndrome scales to better disentangle associations between the ($\Delta$) brain embedding score and emerging mental health conditions during puberty. For example, dimensional approaches may further help to investigate whether the brain embedding scores are sensitive to capture neuronal variance of early pubertal timing (e.g., early menarche in females[44] and their relationship to internalizing psychopathology[45]). To overcome sample size restrictions often observed in clinical samples, leveraging the longitudinal data from ABCD may further help to investigate the marker's sensitivity to capture refined, but biologically meaningful, mental health processes related to brain dynamics[46]. Our validation analyses in the ABCD sample yielded small—yet significant—effect for the psychopathology associations, indicating that our model may also be sensitive to subtle psychopathological manifestations that do not (yet) exceed a clinical threshold. These initial results suggest that our approach may also render useful to study psychopathology in future releases of the ABCD sample.

Potential limitations might stem from the fact that only two imaging modalities, that is brain volume and surface area, were integrated in our framework. Since brain volume and area follow a comparable normative developmental trajectory from late childhood into late adolescence[40], building similarity networks on both measures may result in robust and non-sparse reference space that allows to better disentangle sex effects, since additional heteroscedasticity of different imaging measures may be mitigated[41].

However, beyond the proof-of-concept of the current study, we nevertheless deem it important to extend our approach with additional (imaging) modalities to tests its generalizability beyond the two

imaging features. Furthermore, integrating additive data sources may result in a more holistic (i.e., multimodal) phenotype representing brain development or dynamics which may help to explain additional variance in behavioral or mental health measures and thus may substantiate the brain embedding score utility in capturing brain trajectories. In addition, focusing on more than the first brain embedding might also help to explain additional variance in the tested associations. However, we consider it essential for future work to systematically test how modality-specific information is encoded in the brain embeddings before testing if later embeddings contain biologically meaningful between or within subject variances. Lastly, since our model results in a single brain embedding score, our current approach is limited in its spatial interpretability. While univariate analyses may yield the highest interpretability, they come at the cost of methodological hurdles, such as multicollinearity, high dimensionality, or conflicting feature importance despite similar model performances (e.g.,[47,48]). Beyond those hurdles, modeling brain maturation and sex differences introduces additional variance, which might be difficult to model in an univariate fashion[41,49]. Our approach of integrating high-dimensional data into a single score may facilitate the modeling of developmental slopes and might thus be better suitable for tracking within-subject changes. To address limitations in interpretability, we provided brain maps illustrating the correlation of each brain feature to the brain embedding (Supplementary Fig. 1). Similarity in these maps between cohorts supports robustness of the observed patterns. Other approaches, such as feature deletion[50] may further increase post-hoc interpretability.

We introduced an approach which allows to integrate high high-dimensional imaging data into a coherent feature space in which subjects can be localized by a single brain embedding score. We suggest that transferring this mapping to other datasets results in a new imaging phenotype which inherits a sensitivity to capture meaningful and biologically relevant processes shaping human brain dynamics.

## Methods

### Sample descriptions

**PNC**. As source model we used imaging data from the Philadelphia Neurodevelopmental Cohort (PNC), a large-scale cross-sectional population study of child and youth between 8- and 21-years age dedicated to study (brain) development. All PNC study procedures were approved by institutional review boards of the University of Pennsylvania and the Children's Hospital of Philadelphia. All participants or their caregiver provided written informed consent. Data in the PNC sample was acquired from a single site[13]. We included data from $N = 1594$ individuals with available T1-weighted imaging (females = 834, age: M = 14.95, SD = 3.69; sex was obtained from electronical medical records). We used brain area and volume of 68 cortical brain regions matching the Desikan-Killiany atlas[51] estimated from T1 MRI images using FreeSurfer (version 7.1.1)[52].

**ABCD**. The Adolescent Brain Cognitive Development (ABCD) Study is a 10-year longitudinal study of children recruited at age 9 to 10 aiming at characterizing brain developmental trajectories. Overall ~11.000 children were recruited across 21 different sites in the United States[14]. Study procedures have been approved by either the local site Institutional Review Board (IRB) or by local IRB reliance agreements with the central IRB at the University of California. All participants and their parents provided written informed consent. Data for the current study was obtained from ABCD release 4.0 utilizing phenotypic and imaging data from the baseline and 2-years follow-up study visit. Preprocessed imaging data from the Desikan-Killiany atlas (68 regions)[51] were downloaded from the NIMH data archive. Since we were interested in the longitudinal data, we included only children having MRI data from both baseline and 2-years follow-up visit ($N = 7776$, females = 3587,

$age_{baseline}$: M = 9.90, SD = 0.62; $age_{follow-up}$: M = 11.90, SD = 0.65; sex is defined as biological sex assigned at birth).

**HBN.** The Healthy Brain Network (HBN) is a community sample of children and adolescent (ages 5–21) in the New York area aiming at capturing and investigating the heterogeneity in developmental psychopathology and its biological underpinnings[15]. Imaging data was acquired across four different scanning sites and study procedures were approved by the Chesapeake IRB. All participants or their caregiver provided written informed consent. Brain area and volume of 68 cortical brain regions from T1 MRI images were estimated according to the Desikan-Killiany atlas[51] using FreeSurfer (version 7.1.1)[52]. Based on clinical diagnostic information and the presence of a primary diagnosis, we integrated data from N = 2271 (females = 784, age: M = 10.43, SD = 3.45; sex is defined as biological sex assigned at birth) participants.

## Model building and testing
Brain volume and area from the Desikan-Killiany atlas[51] were used to construct fused similarity networks with *snfpy* (version 0.2.2, https://github.com/rmarkello/snfpy). In the following we will briefly describe the SNF workflow but refer the reader to Wang et al.[8] for a more detailed description: First, we generated subject x subject affinity networks for MRI area and volume by converting between-subject (squared euclidean) distances to similarities with a scaled exponential kernel, respectively. Next, SNF iteratively fused each feature affinity matrix resulting into one symmetric similarity matrix integrating information from all data sources. Both previous steps are governed by the hyperparameters $K$ (i.e., the number of neighbors to consider) and $\mu$ (i.e., weighting of between subjects' edges) with $K \in [1, 2, ..., i]$, $i \in \mathbb{Z}$ and $\mu \in \mathbb{R}^+$. Markello et al.[6] performed a grid-search across 10.000 hyperparameter combinations and reported consistent embeddings across all combinations ($r_{mean}$ = 0.97), suggesting a neglectable effect of extensive hyperparameter tuning for consecutive analyses aiming at continuous representations. We thus set $K = 30$ and $\mu = 0.8$ in accordance with the suggested range of values in *snfpy*. The fused matrix is full rank and can then be either subjected to clustering or dimensionality reduction to achieve continuous representation of the data in a low-dimensional space. Since we were interested in the latter, we performed diffusion map embedding on the fused network to derive low-dimensional representations of the imaging data using *BrainSpace* (version 0.1.3)[53]. Diffusion map embedding is a non-linear dimensionality reduction technique that projects the raw data onto dimensions (i.e., brain embeddings) that encode the primary axes of between-subject similarity. The resulting embeddings are unitless and subjects can be localized according to their inter-subject similarity along these dimensions[6]. Critically, diffusion map embedding has been shown to be sensitive to non-linear relationships and robust against noise perturbations compared to other techniques, such as Principal Component Analyses (PCA)[10,54]. The diffusion time parameter $t$ was set to zero to model the most global relationship of the input data[10]. For further analyses the first brain embedding was used, as it captures the highest variance akin to PCA.

For our machine learning framework we then trained an Elastic Net in scikit-learn (version 1.0.2)[55] to learn the mappings between the raw feature space (i.e., area and volume MRI data, each with shape 1594 × 34 after averaging features across both hemispheres) and the first brain embedding. Since our goal was to maximize out of sample generalizability, we (1) trained the model with default parameters (l1_ratio = 0.5 balancing L1 and L2 norm regularization, alpha= 1.0 which tunes the overall penalty strength) aiming at minimizing overfitting to the training set and (2) utilized an instance-based supervised domain adaptation (Transfer AdaBoost for Regression; TrAdaBoostR2)[28] implemented in ADAPT (version 0.4.1)[56].

TrAdaBoostR2 combines a source (PNC) and target data set into a single set and performs reverse boosting in which weights of the source instances poorly predicted decrease at each iteration while the ones of the target instances increases, thus shifting the relative importance towards the target set[28]. Thus, the algorithm makes use of those source instances that are similar to the target domain and ignores the ones that are more dissimilar. Since increasing the boosting iterations may lead to overfitting, the algorithm per default uses the weighted median of the last N/2 iterations for prediction. To avoid data leakage, we used held-out data from the ABCD and HBN: For the ABCD data we used N = 3984 (females = 2027, age: M = 9.95, SD = 0.63) children for which only baseline imaging data was available at release 4.0. In the HBN sample we used imaging data from a healthy sample of N = 389 (females = 162, age: M = 10.45, SD = 3.81) for which no primary diagnosis was reported. Of note, for the latter we did pool subjects with the label 'no diagnoses' either based on a complete or aborted evaluation. For both datasets we used brain volume and area from the Desikan-Killiany atlas[51]. Since MRI data was acquired on different scanners both for the ABCD and HBN data, we harmonized both the volume and area imaging data individually using *neuroCombat* (version 0.2.12)[57]. Of note, batch correction was performed on individual timepoints for the ABCD data and separated for train and test set.

After fitting with domain adaptation, we applied the model to unseen test data from the ABCD and HBN, respectively. To quantify the quality of predictions we additionally also performed SNF and diffusion map embedding on the ABCD and HBN test sample and calculated error metrics (MSE; MAE; RMSE) and $R^2$ and correlation values between the predicted and 'true' first brain embedding after orthogonal Procrustes alignment with *mapalign* (version 0.3.0, https://github.com/satra/mapalign)[58]. A schematic representation of the workflow is depicted in Fig. 1A. In addition, Supplementary Fig. 1 depict the correlation between the raw features and the first brain embedding in the HBN and ABCD sample, respectively.

## Modeling puberty
Pubertal development in the ABCD and HBN sample was assessed with the Pubertal Development Scale (PDS) which was designed to resemble the Tanner stages without the need of a physical examination[30,38]. The child's pubertal development is rated on a four-point Likert scale ranging from 'has not begun' to 'completed' with one exception, that is a binary response item regarding females' menarche. Overall, there are general and sex-specific items that are administered with respect to the biological sex, e.g., voice-deepening or breast development. The rating can be conducted by the children or their caregivers, thus reflecting self or other-perceived pubertal maturation. In the ABCD study both children and caregiver report are available for both timepoints[38], whereas in the HBN study only participant responses are available[15]. Individual sex-specific item scores were used to calculate the average PDS score ($PDS_{mean}$) in line with procedure described in Herting et al.[37]. For longitudinal associations, we additionally calculated a $\Delta$ PDS score as a marker for pubertal maturation, that is the difference between baseline and 2-years follow-up PDS score. Moreover, pubertal category scores were derived for males and females. For males, the sum of three items related to pubic and facial hair growth as well as voice deepening was calculated. For females, pubic hair growth and breast development was summed and information about the menarche was additionally incorporated. Eventually, pubertal scores were converted into pubertal categories ranging from prepubertal to post pubertal based on the ABCD conversion scheme (see Supplementary Table 1). The frequency of pubertal categories for the baseline and follow-up data is shown in Supplementary Table 2. $PDS_{mean}$ scores were also calculated in the HBN sample to test for out-of-sample replicability and generalizability.

## Modeling psychopathology

In the HBN sample each participant and their caregiver underwent an online version of a semi-structured DSM-5 based psychiatric interview (K-SADS)[59] to derive clinical diagnoses. Consensus diagnoses for each participant are made based on the overlap of the child and caregiver interview by a research clinician[15]. We calculated the sum of all consensus diagnoses per subject as a proxy for psychopathology severity. Frequencies of psychopathology measures can be derived from Supplementary Table 3.

## Association analyses

All association analyses were performed with *statsmodels* (version 0.13.2)[60]. To test for associations between pubertal development and the predicted brain embeddings in the ABCD study, we implemented linear models for each timepoint (i.e., baseline and 2-year follow-up) with the respective brain embedding as dependent variable (DV) and the $PDS_{mean}$ score as independent variable (IV) with two-sided significance testing. For all associations we additionally report partial-eta-squared ($\eta^2$) per predictor of interest. Since we were particularly interested whether the difference between both brain embeddings ($\Delta$ brain embedding) captures biological variance that is associated to brain dynamics, we performed an additional linear model with $\Delta$ brain embedding as DV and the $\Delta$ $PDS_{mean}$ score as IV. Analyses were stratified for sex and youth and caregiver reports accounting for differences how pubertal development might be perceived[38]. Despite the rather narrow age range at each study visit, age or $\Delta$ age (i.e., the difference in age between baseline and 2 years follow-up accounting for variance in between-visit durations) was added as a covariate to the linear model in ABCD, to rule out that putative pubertal effects merely represent aging effects. For the ABCD sample the number of observations varies between models as the amount of missing data is different per timepoint and depends on whether the participants themselves or their caregiver provided the data. In the HBN sample, we tested the association between the predicted brain embedding (DV) and the sum of diagnoses (IV), which we introduced as a proxy for psychopathology severity. Based on the close relationship between puberty and emerging mental disorders, we additionally calculated linear models which included both the summed diagnoses and the $PDS_{mean}$ score as IVs and one model containing an interaction term summed diagnoses: $PDS_{mean}$ score next to the main effects. $PDS_{mean}$ score was based on participant reports. For the HBN sample the number of observations varies between models as missing data was excluded on a model-by-model bases, i.e., dependent on the IVs of interest. Linear models were stratified for sex and age and site were added as covariates of no interest. All linear models were Bonferroni corrected for multiple comparisons[61].

## Reporting summary

Further information on research design is available in the Nature Portfolio Reporting Summary linked to this article.

## Data availability

Data used in this study was accessed under data use agreements of the respective study cohorts (ABCD, PNC, HBN). Raw data must not be shared directly by the study authors, but researchers can get access through own data use agreement and use our shared scripts to reproduce the results.

## Code availability

All code used in this manuscript is available on github (https://github.com/dominikkraft/DomAdapt_BrainNetFusion) and Zenodo (https://doi.org/10.5281/zenodo.8223987; release v1.0.0) and builds upon python 3.7.11. Basic data handling relied on *pandas* (version 1.3.5)[62] and *numpy* (version 1.21.5)[63]. Data visualization relied on matplotlib (version 3.5.1)[64] and seaborn (version 0.11.2)[65].

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

## Acknowledgements

This work was funded by the Research Council of Norway (#323961, T.K.) and the South-Eastern Norway Regional Health Authority (#2019107, #2020086, D.A.). T.K. is a member of the Machine Learning Cluster of Excellence, EXC number 2064/1 – Project number 39072764. This work was supported by the BMBF-funded de.NBI Cloud within the German Network for Bioinformatics Infrastructure (de.NBI) (031A537B, 031A533A, 031A538A, 031A533B, 031A535A, 031A537C, 031A534A, 031A532B). The authors used data from the Philadelphia Neurodevelopmental Cohort (PNC, access permission no 29782), the Adolescent Brain Cognitive Development$^{SM}$ Study (ABCD, abcdstudy.org), and the Healthy Brain Networks (HBN, data.healthybrainnetwork.org). Support for the collection of the PNC data set was provided by grant #RC2MH089983 awarded to Raquel Gur, MD, PhD, and #RC2MH089924 awarded to Hakon Hakonarson, MD, PhD. ABCD data, held in the NIMH Data Archive (NDA), is a multisite, longitudinal study designed to recruit more than 10,000 children age 9–10 and follow them over 10 years into early adulthood. The ABCD Study® is supported by the National Institutes of Health and additional federal partners under award numbers U01DA041048, U01DA050989, U01DA051016, U01DA041022, U01DA051018, U01DA051037, U01DA050987, U01DA041174, U01DA041106, U01DA041117, U01DA041028, U01DA041134, U01DA050988, U01DA051039, U01DA041156, U01DA041025, U01DA041120, U01DA051038, U01DA041148, U01DA041093, U01DA041089, U24DA041123, U24DA041147. A full list of supporters is available at https://abcdstudy.org/federal-partners.html. A listing of participating sites and a complete listing of the study investigators can be found at https://abcdstudy.org/consortium_members/. PNC, HBN, and ABCD consortium investigators designed and implemented the respective studies and/or provided data but did not participate in the analysis or writing of this report. This manuscript reflects the views of the authors and does not necessarily reflect the opinions or views of any other agency, organization, employer or company.

## Author contributions

D.K.: Conceptualization; Data curation; Formal analysis; Investigation; Methodology; Project administration; Software; Visualization; Writing—original draft; Writing—review, editing, and approval of the paper. D.A.: Data curation, Writing—review, editing and approval of the paper. T.K.: Conceptualization; Project administration; Methodology, Funding acquisition, Writing—original draft; Writing—review, editing, and approval of the paper.

## Competing interests

The authors declare no competing interests.
