## [Peer Review File · Nature Communications]

Domain adapted brain network fusion captures variance related to pubertal brain development and mental healthReviewer #1 (Remarks to the Author):

In this manuscript, Kraft et al. present exciting results demonstrating that similarity network fusion can be applied to generate diffusion map embeddings that subsequently exhibit associations with key developmental and psychiatric variables. Strengths of this work include the use of several different developmental datasets (use of PNC to generate initial embeddings followed by validation in ABCD and HBN) and the validation using pubertal scores and psychiatric diagnoses. The results presented here are impressive and provide solid evidence that this approach may reveal meaningful associations between brain and behavior in healthy and clinical populations. However, I do have some general concerns about the work as it is currently presented. Please find below my comments and questions.

1. This approach does not seem to support interpretability of any sort that allows us to capture the underlying brain features that are driving this relationship. In other words, there's no way to know from these analyses, if (or how) specific brain regions (and their respective surface area and/or volume) are responsible for the associations with pubertal development or psychopathology. Although interpretability is by no means a requirement, I am curious about how the authors envision such a technique playing a role in research (and clinical practice) moving forward. A brief comment on this in the discussion would be helpful.
3. Regarding the machine framework to develop models to predict embeddings in unseen samples: Was any cross-validation used? What were the "default parameters" used?
4. It's interesting that the difference between the embeddings begins to diverge from the 'mid-pubertal' stage onwards and become more pronounced after menarche across sexes - can authors comment on why this might be the case in the discussion? Are there key structural changes in the brain that could be driving this?
5. How might authors explain the discrepancy in associations for the pubertal development scale for caregiver reports compared to youth self-reports?
6. Some information about how the summarised pubertal scores were converted into pubertal categories would be helpful.
7. Authors state that the predicted embedding score exhibits association with psychopathology in females but not males in HBN, what might explain this?
8. Authors mention they only used the first embedding as it captures the highest variance. However, given that they are computing associations with puberty and psychopathology, both of which are likely to exhibit a decent amount of variability at the individual level, can the authors comment on how integrating additional embeddings in future analyses may affect the results?
9. Was psychopathology considered in ABCD at all? Were subjects with diagnoses excluded from the analyses? Did authors consider using participants with known diagnoses in ABCD to validate their results regarding the psychopathology associations obtained in HBN?
10. Please reference Figure 1B in the text and provide some context for it.

Reviewer #2 (Remarks to the Author):

A great big data project. The authors present a relatively new multi-dimensional data reductions technique and then used a supervised domain adaption to make it transferable to other datasets. The study is commendable on its approach, and provides a valuable contribution. However, I have a few suggestions and concerns that should be addressed to improve the manuscript.

-Firstly, I acknowledge that it is a very technical approach, but throughout the whole paper there is room for improvement in the writing being more accessible for a general audience.

-I'm not convinced of the use of the term "signatures of pubertal brain development" used in the title. The study simply found an embedding from multidimensional data that captured some variance in puberty that differed in only females in only late puberty. That is not to say the finding is not worthwhile, but does not quite capture the comprehensive and specific characterisation typically associated with a 'signature'.

- The introduction content is just about the method, and does not raise any literature around what is known about brain development associated with puberty or with psychopathology.
 - It is great the models were tested in independent datasets. The study used three cohorts (PNC, ABCD, HBN): PNC for training of 'typical brains', ABCD for examining puberty and HBN for examining psychopathology. However, each of the three cohorts used, all have measures of puberty and psychopathology. I don't understand why there was not an attempt to replicate across cohorts.
 - I would be interested to hear why the authors chose to use surface area and volume, rather than surface area and cortical thickness (which are more typically used) or even all three. Volume is just a function of surface area and thickness, so will be highly colinear with surface area.
 - The training dataset has quite a large age range. Did you examine to what extent the first embedding is related to age (and overall brain size)? Was any attempt made to exclude psychopathology from the training dataset in PNC?
 - Would it be possible to visualise the brain features (regional volume and surface area) associated with the first embedding? And that captured the most variance between individuals and over age (in the normative model), and associated with both puberty and psychopathology?
 - When investigating puberty, there is established literature (even specifically in the ABCD cohort) around the need to covary for BMI, SES and race. This was not addressed or mentioned.
 - I am not convinced that the approach taken to examine psychopathology is the most appropriate or interpretable. The authors use the total number of comorbid diagnoses as the marker of severity. This range is 1-10 comorbid diagnoses (mean 2.1, sd 1.6), so very unlikely individuals have more than 3-4 diagnoses. It also suggests for example, an individual with severe schizophrenia is not as impacted and an individual that only just meets criteria of ADHD and depression). Whilst the authors do acknowledge that it is a coarse measure of psychopathology, even without looking at different diagnoses separately, the cohorts have broad spectrum dimensional measures (such as the CBCL), that could not only be used across the entire cohort (typical and disorders), but would also provide greater variance in the psychopathology measure. At the very least a histogram should be provided with a) the number of different comorbidities (your unit of psychopathology severity), and b) one showing the prevalence of each disorder in the sample.
- Minor
- In the results it would be good to note what the change in PDS scores were for both males and females.
 - The sentence bottom page 5 is not clear and should be re-phrased. Also in that sentence, 'adolescent' should be 'adolescence'.

Reviewer #3 (Remarks to the Author):

This is an interesting paper reporting on a novel approach in a proof of concept analysis. The approach combines SNF to derive similarity across subjects (embeddings) from an initial dataset (PNC), then uses machine learning model to train for prediction of brain embeddings based on two raw MRI features (cortical area, and volume) using both the original dataset and two subsets from unseen datasets (ABCD, HBN). The model was then validated on data from ABCD and HBN that was not used in the training phase to derive two predictions for ABCD longitudinal data (baseline, followup) and one prediction of brain embeddings for HBN data. Finally, two independent variables associated with brain development/health were tested to see if they predicted brain embeddings derived from SNF analysis: (i) pubertal development using longitudinal data from ABCD, and (ii)

sum of diagnoses (a proxy of psychopathological severity) from HBN. The authors report that the model achieved high performance in unseen data, and that brain embeddings derived from unseen data were associated with meaningful variables (PDS and clinical diagnoses in females). The paper is well written, proof of potential utility is shown in the current paper and limitations of the data used and aspects of the approach are clearly laid out. I have the following questions/suggestions:

Comments/questions:

-what was the quality control approach used across datasets? were there key differences in data quality between ABCD and HBN datasets that may contribute to poorer model performance in HBN vs ABCD data? Did data quality in HBN also correlate with diagnostic sum (measure of psychopathology)?

-I found the description of analyses examining associations bw brain embeddings and pubertal/clinical variables somewhat confusing at times. It would be helpful to use brain embeddings consistently when referring to embeddings to clarify for reader.

-It is notable that effect sizes for associations between brain embeddings and caregiver pubertal score are small-moderate for baseline data, small for 2 year data and very small for change scores (and results are weaker for youth report data). Given effect sizes are small, this method may require very large datasets for both training and testing which may limit applicability to clinical samples where large datasets with extensive characterization are limited.

-A table providing some information about the data used in the current study across datasets would be useful, this could be supplemental. Need a table for PNC, HBN and ABCD with information on %female and PDS scores/%F and M in different categories as well as information about no of diagnoses for M and F participants to better understand findings that were significant in F but not M in HBN dataset.

Response to Reviewers

Content

Response to all Reviewers	p. 1
Response to Reviewer #1	p. 2ff.
Response to Reviewer #2	p. 15ff.
Response to Reviewer #3	p. 30ff.

Response to all Reviewers:

We would like to thank the reviewers for their constructive feedback which clearly helped improving our manuscript. In brief, our major revisions include:

- We have integrated additional validation analyses. While we previously used the ABCD sample to test for puberty associations and the HBN samples to test for psychopathology associations, we now test both, puberty, and psychopathology associations, in both cohorts. These analyses further support our initial findings.
- We have integrated additional information to characterize psychopathology in the HBN sample, using both diagnosis data and dimensional data from the Child Behavior Checklist. We observe similar effects for the newly integrated dimensional measure as previously observed when using number of diagnoses as a measure of severity.
- We have complemented our manuscript with six additional figures and five additional tables that provide further insights. Of note, this includes figures showing the relationship between raw feature and the first brain embedding, allowing us to contextualize the observed associations with brain embeddings to the underlying neuroanatomical source.

We have addressed all reviewer comments and provide individual responses below in *italics* font. For convenience, we incorporated resulting changes to the manuscript directly in the responses in quotation marks. Major changes are also highlighted with **green** color in our revised manuscript.

Reviewer #1:

In this manuscript, Kraft et al. present exciting results demonstrating that similarity network fusion can be applied to generate diffusion map embeddings that subsequently exhibit associations with key developmental and psychiatric variables. Strengths of this work include the use of several different developmental datasets (use of PNC to generate initial embeddings followed by validation in ABCD and HBN) and the validation using pubertal scores and psychiatric diagnoses. The results presented here are impressive and provide solid evidence that this approach may reveal meaningful associations between brain and behavior in healthy and clinical populations. However, I do have some general concerns about the work as it is currently presented. Please find below my comments and questions.

We would like to thank the reviewer for the positive feedback and fruitful comments.

1. This approach does not seem to support interpretability of any sort that allows us to capture the underlying brain features that are driving this relationship. In other words, there's no way to know from these analyses, if (or how) specific brain regions (and their respective surface area and/or volume) are responsible for the associations with pubertal development or psychopathology. Although interpretability is by no means a requirement, I am curious about how the authors envision such a technique playing a role in research (and clinical practice) moving forward. A brief comment on this in the discussion would be helpful.

We agree with the reviewer that the benefit of high dimensional data fusion comes at the cost of limitations in spatial interpretability. Below, we first recapitulate the benefit of dimension reduction, and argue that it outweighs the limitations in applications such as the one deployed here. We then provide solutions how the limitations of spatial interpretability can be overcome. We added an analysis showing the contribution of brain regions to the brain embeddings and discuss how further approaches could be deployed in future studies.

While the raw spatial resolution (e.g., univariate ROI or voxel level) may yield the highest interpretability, the high dimensionality comes with certain disadvantages. First, the data is often approached with univariate approaches where an appropriate correction for multiple comparisons needs to be implemented (Rosenberg and Finn, 2022; DOI: 10.1038/s41593-022-01110-9). Moreover, multicollinearity between features might additionally hamper feature importance calculations and its interpretation. In multivariate settings, different machine learning models may yield predictions with similar model performance but with different or even conflicting explanations on the feature level (Del Giudice, 2021; DOI: 10.31234/osf.io/4vq8f). Beyond those methodological hurdles, a recent study by Bottenhorn et al. (2022; DOI: 10.1101/2022.12.19.521089) reported a high level of intra- and interindividual variation in brain maturation across regions and measures. Integrating additional variables of interest such as sex adds an additional layer of variance complexity (see e.g., Wierenga et al, 2019; DOI: 10.1162/jocn_a_01375), making it a difficult equation to model.

Our approach of integrating high dimensional data into a single fusion score may help to overcome these difficulties by unifying different sources of variance and thus may offer an easier approach to map developmental trajectories or slopes. Even though our framework in its current form is far from clinical application, we consider its strength of integrating high-

dimensional multimodal data into a lower dimension potentially interesting for a clinical application, since (1) it may reveal relevant patterns (e.g. in subgroups) that are not detectable in a univariate fashion, (2) put subjects in the context of a reference population, and (3) delivers single brain scores that can be traced across multiple repeated scans to quantify multidimensional longitudinal changes in one number.

However, as several applications may indeed require interpretability in brain space, we want to note that certain post-hoc analyses may target interpretability of the brain embeddings. The most straightforward approach might be to correlate the brain embeddings with the raw features and derive brain maps that give an indicator of feature contribution. In this revision, we have performed such an analysis and now provide the relevant brain maps in the corresponding new Supplementary Figure 1, for convenience pasted below. We want to highlight, that there are several other approaches that may deliver further insights into feature interpretability, such as for example iteratively deleting features and evaluating the effect on model performance and prediction estimates (Scheinost et al., 2019; DOI: 10.1016/j.neuroimage.2019.02.057). For doing so, we would nevertheless consider it important to systematically evaluate such approaches for our framework in a dedicated study.

As suggested by the reviewer, we now also discuss the challenges of spatial interpretability in more detail in the ‘Limitations and Future Directions’ paragraph (p.16, l. 446ff.):

“Lastly, since our model results in a single brain embedding score, our current approach is limited in its spatial interpretability. While univariate analyses may yield the highest interpretability, they come at the cost of methodological hurdles, such as multicollinearity, high dimensionality, or conflicting feature importance despite similar model performances (e.g.,^{47,48}). Beyond those hurdles, modeling brain maturation and sex differences introduces additional variance, which might be difficult to model in an univariate fashion^{41,49}. Our approach of integrating high-dimensional data into a single score may facilitate the modeling of developmental slopes and might thus be better suitable for tracking within-subject changes. To address limitations in interpretability, we provided brain maps illustrating the correlation of each brain feature to the brain embedding (Supplementary Fig. 1). Similarity in these maps between cohorts supports robustness of the observed patterns. Other approaches, such as feature deletion⁵⁰ may further increase post-hoc interpretability.”

41. Bottenhorn, K. L., Cardenas-Iniguez, C., Mills, K. L., Laird, A. R. & Herting, M. M. *Profiling intra- and inter-individual differences in child and adolescent brain development*. <http://biorxiv.org/lookup/doi/10.1101/2022.12.19.521089> (2022) doi:10.1101/2022.12.19.521089.
47. Rosenberg, M. D. & Finn, E. S. How to establish robust brain–behavior relationships without thousands of individuals. *Nat. Neurosci.* 25, 835–837 (2022).
48. Del Giudice, M. *The Prediction-Explanation Fallacy: A Pervasive Problem in Scientific Applications of Machine Learning*. <https://osf.io/4vq8f> (2021) doi:10.31234/osf.io/4vq8f.
49. Wierenga, L. M., Bos, M. G. N., Van Rossenberg, F. & Crone, E. A. Sex Effects on Development of Brain Structure and Executive Functions: Greater Variance than Mean Effects. *J. Cogn. Neurosci.* 31, 730–753 (2019).
50. Scheinost, D. et al. Ten simple rules for predictive modeling of individual differences in neuroimaging. *NeuroImage* 193, 35–45 (2019).

Furthermore, we added Supplementary Figure 1, for convenience pasted below:

Supplementary Fig. 1: Correlation between raw brain features and first brain embedding. Correlation values indicate Pearson correlation coefficients. Visualization was performed with *ggseg* (version 1.6.5¹) in R (version 4.2.3). Since algebraic signs of the embeddings might be positive or negative (i.e., akin to an eigenvector), we multiplied negative correlations by (-1) to derive at comparable patterns across cohorts. Spin permutation testing with 5000 iterations indicated significant correlations between brain maps of the ABCD and HBN dataset (all $r > 0.64$, all $p < 0.0004$). Permutations were performed using the ENIGMA Toolbox (version 2.0.3,²).

Supplementary References:

1. Mowinckel, A. M. & Vidal-Piñeiro, D. Visualization of Brain Statistics With R Packages *ggseg* and *ggseg3d*. *Adv. Methods Pract. Psychol. Sci.* 3, 466–483 (2020).
2. Larivière, S. *et al.* The ENIGMA Toolbox: multiscale neural contextualization of multisite neuroimaging datasets. *Nat. Methods* 18, 698–700 (2021).

3. Regarding the machine framework to develop models to predict embeddings in unseen samples: Was any cross-validation used? What were the “default parameters” used?

The machine learning performed in this study contains two steps. (1) training with iterative boosting, and (2) testing in held-out data. We describe each step and the parameters in more detail below.

- (1) Our machine learning framework contains a supervised domain adaptation procedure. The objective of this technique is that it is easier to learn a prediction task in a certain domain when integrating knowledge from a related domain into the learning, such as for example, learning to ride a mountain bike by making use of previous expertise riding a city bike. In neuroimaging, we often deal with small samples that make it difficult to train machine learning models in this limited data without a risk of model failure. Using other, larger samples to learn the same task and then adapting the model to the specific characteristics of the small sample – domain adaptation – can help overcome this issue. Specifically, we combine source data (i.e., PNC) and a subset of the target data (i.e., either of the ABCD or HBN) into a training set to learn the mapping between the raw brain features and the first brain embedding. Domain adaptation is an iterative “reverse boosting” process where in each of ten iterations the algorithm performs a reweighting of the poorly predicted source and target instances such that source instance weights are iteratively decreased and target instance weights increased. The algorithm makes use of those source instances that are similar to the target domain and “ignores” the ones that are more dissimilar. Since increasing the boosting iterations may lead to overfitting, the algorithm per default uses the weighted median of the last $N/2$ iterations for the prediction. Taken together, domain adaptation is a process in which the relative importance shifts towards the target data with the ultimate goal to build a good model for the target data.*
- (2) The resulting model was then validated in held-out datasets in the ABCD and HBN sample, respectively. Of note, this idea resembles the general idea of a hold-out cross validation procedure to test the generalizability of the model. Our train / test splitting was not performed in a standard random split fashion (e.g., by splitting 80:20 into train and test), but rather followed a data maximization rationale. Specifically, we fed all data that could not be used in the association analyses (e.g., due to missing imaging data at follow-up) into the domain adaptation training set and used all data without missingness as the test set. Of course, a ‘classical’ random train - test split would also be possible, but under the premise of reduced statistical power in the subsequent association analyses.*

As furthermore stated in the manuscript we aimed at minimizing overfitting (which could for example happen when the source dataset is larger than the target data, e.g., PNC vs HBN) and maximizing generalizability and thus did not perform any hyper-parameter tuning which would have required an additional (nested) cross-validation scheme. Instead, we used the default parameters for the Elastic Net, which refer to the two default parameters used in the

sklearn.linear_model.ElasticNet implementation: mixing parameter l1_ratio = 0.5 which balances L1 and L2 norm regularization and the tuning parameter alpha = 1.0 which controls the overall strength of the penalty.

We have now added this information to the method section (p. 19, l. 546ff.):

“Since our goal was to maximize out of sample generalizability, we 1) trained the model with default parameters (l1_ratio = 0.5 balancing L1 and L2 norm regularization, alpha= 1.0 which tunes the overall penalty strength) aiming at minimizing overfitting to the training set and [..]”

Additionally, we now also include additional information for the domain adaptation to increase clarity for the reader (p. 19, l. 551ff.):

“[...] TrAdaBoostR2 combines a source (PNC) and target data set into a single set and performs reverse boosting in which weights of the source instances poorly predicted decrease at each iteration while the ones of the target instances increases, thus shifting the relative importance towards the target set²⁸. Thus, the algorithm makes use of those source instances that are similar to the target domain and “ignores” the ones that are more dissimilar. Since increasing the boosting iterations may lead to overfitting, the algorithm per default uses the weighted median of the last N/2 iterations for prediction.”

28. Pardoe, D. & Stone, P. Boosting for Regression Transfer. in *Proceedings of the 27th International Conference on International Conference on Machine Learning* 863–870 (Omnipress, 2010).

4. It's interesting that the difference between the embeddings begins to diverge from the 'mid-pubertal' stage onwards and become more pronounced after menarche across sexes - can authors comment on why this might be the case in the discussion? Are there key structural changes in the brain that could be driving this?

Given the narrow age range in the ABCD study, pubertal categories may serve as a proxy for pubertal timing in a sense that participants in later pubertal stages (e.g., late pubertal) underwent earlier pubertal maturation compared to their peers at a given time point. In particular, it has been shown that a greater (pubertal) stage-for-a-given-age is related to smaller / thinner cortices, given that normative brain developmental trajectories are typically described by decreasing gray matter metrics (e.g., Dehestani et al., 2023, DOI: 10.1016/j.dcn.2023.101227). These effects however appear to be rather distributed across the whole brain instead of being limited to particular cortical or subcortical regions (cf. Vijayakumar et al., 2018; DOI: 10.1016/j.neubiorev.2018.06.004). A recent study deploying a brain age framework investigating the relationship between 'puberty age gap' and 'brain age gap', also indicates a more mature brain structure in subjects undergoing earlier puberty (Dehestani et al., 2023; DOI: 10.1016/j.dcn.2023.101227). Our results provide further ground to these findings, suggesting that larger divergence in delta brain embeddings may reflect a more mature brain in females that show greater stage-for-a-given-age. As noted in the manuscript, the current ABCD data release only contains few male participants at later pubertal stages (see also our response to Reviewer 3, comment 4 which contains a Table showing the total number and percentages of participants per pubertal stages) and thus limits

our ability to fully investigate the effects observed for the delta brain embedding in males. Future releases may thus help to further test the putative mechanism described above.

We now include this putative explanation in the respective section of the discussion (p. 13, l.358ff.):

“Moreover, it appears that the Δ brain embeddings for both sexes follow a comparable distribution in early pubertal stages, whereas from females’ menarche onwards, both patterns start to deviate from each other. Given the narrow age range in the ABCD study, pubertal categories may serve as a proxy for pubertal timing with females often undergoing earlier puberty. Greater pubertal stage for a given age has been related to more mature, i.e., thinner cortices (e.g. ²²), which might be a putative explanation for the divergence in the Δ brain embedding. Upcoming releases of the ABCD data may help to further investigate the Δ brain embedding and its ability to capture subtle biological processes like pubertal maturation.”

22. Dehestani, N., Whittle, S., Vijayakumar, N. & Silk, T. J. Developmental brain changes during puberty and associations with mental health problems. *Dev. Cogn. Neurosci.* 60, 101227 (2023).

5. How might authors explain the discrepancy in associations for the pubertal development scale for caregiver reports compared to youth self-reports?

For the associations between brain embeddings and puberty scores (PDS), we observed discrepancies only for the baseline associations, where we found associations for the caregiver but not the youth reports (see Figure 2 in main manuscript). We deem it important to reconsider that the PDS scale youth report refers to a self-report about perceived pubertal development and might thus be biased by the subjective impression the rater has about his/her own pubertal development (cf. Herting et al., 2020; DOI: 10.3389/fendo.2020.549928). In the discussion of our submitted manuscript, we previously described the issue as rating one’s own pubertal maturation (e.g., has your voice changed) might be difficult at these early ages (mean age at baseline was 9.9 years in our ABCD sample), leading to biased self-reports. We however acknowledge that the opposite, that is an over- or underestimation of pubertal development, on the caregiver side, might be also a putative explanation (Herting et al., 2020).

Since we cannot ultimately disentangle which of these scenarios might have driven the association results, we now decided to substantiate the discussion with Supplementary Figure 5, which contrasts caregiver and youth PDS mean distributions across both timepoints, to provide the reader with additional information. Of note, the kernel density plots (panel A-D) show both the caregiver (solid line) and youth (dashed line) reports for the same subjects, as it allows for a better comparison between ratings and to calculate the correlations between both ratings. We have also added a Supplementary Table 4, which depicts the total number of subjects / percentage of subjects per pubertal category stratified for sex and self-report responder to the manuscript, providing additional information to the reader

Supplementary Fig. 5: Distribution of puberty ratings in the ABCD sample stratified for sex, timepoints, and respondent (Panel A-D). Solid lines indicate caregiver and dashed line indicate youth reports. Higher correlation values between puberty ratings at follow-up indicate better alignment between reports from different sources. Panel E and F refer to changes in puberty scores (i.e., Δ PDS) for males and females. All plots only show data from participants for which data from both timepoints was available. r = Pearson correlation coefficient.

For males at baseline (Panel B) the plot shows a clear divergence between the caregiver and youth report with the latter being shifted towards higher values, indicating higher perceived pubertal maturation in youth. This difference gets smaller in the two-year follow-up data (D). This is also substantiated by the correlation coefficient which almost doubles between both timepoints. Since it is well known that boys pubertal timing is a bit later, it might be true for the male participants at baseline that it is particularly difficult to correctly rate their own pubertal maturation, such as voice changes.

For females we also observe an increasing alignment between both reports at the two-year follow-up, however the changes are not as pronounced as in males. Of note, while we observed discrepancies in the association results in female baseline data, it is important to note that these discrepancies appear only minor, since the identified p -value of approximately 0.005 just misses the Bonferroni adjusted alpha level of 0.004. The absence of the effect in youth reports data might therefore be also driven by the strict correction for multiple comparisons.

We have now incorporated this additional information and a broader discussion in the manuscript (p.12, l. 328ff.):

“[...] we observed significant cross-sectional associations between the predicted brain embedding scores and puberty measures for all models at all timepoints except for baseline data based on youth reports, which might have been biased by the difficulty to rate one’s own pubertal maturation at these early ages. Such bias appears to be particularly true for males at baseline (see Supplementary Fig. 5). In addition, we observed higher correlations between

caregiver and youth reports for the 2-year follow-up, suggesting an overall better alignment between reports, potentially minimizing biases.”

6. Some information about how the summarised pubertal scores were converted into pubertal categories would be helpful.

Pubertal category scores were converted into pubertal categories following a conversion scheme provided by the ABCD variables ‘pds_p_ss_female_category’ and ‘pds_p_ss_male_category’ for females and males respectively. We now include this conversion scheme in the Supplemental Material and refer to it in the main text (p.21, l.598ff.):

“Eventually, pubertal scores were converted into pubertal categories ranging from prepubertal to post pubertal based on the ABCD conversion scheme (see Supplementary Table 3).”

Supplementary Table 3: Pubertal category conversion scheme

sex	conversion scheme	pubertal category
female	premenarcheal + pubertal score = 2	prepubertal
	premenarcheal + pubertal score = 3	early pubertal
	premenarcheal + pubertal score >= 3	mid pubertal
	postmenarcheal + pubertal score <= 7	late pubertal
	postmenarcheal + pubertal score = 8	postpubertal
male	pubertal score = 3	prepubertal
	pubertal score >= 4 and <=5	early pubertal
	pubertal score >= 6 and <=8	mid pubertal
	pubertal score >= 9 and <=11	late pubertal
	pubertal score = 12	postpubertal

Note: Pubertal score refers to the Pubertal Category Score, which depicts the sum of three/two PDS items for male and female, respectively. Male: pubic + facial hair growth + voice deepening. Female: pubic hair growth + breast development. Conversion follows the ABCD variables ‘pds_p_ss_female_category’ and ‘pds_p_ss_male_category’.

7. Authors state that the predicted embedding score exhibits association with psychopathology in females but not males in HBN, what might explain this?

When looking at differences in the diagnosis data for males and females in the HBN test sample (figure below), we can observe that whereas the number of diagnosis appears overall higher in males (panel A), the distributions within males and females are similar (panel B), i.e. weighting the overall count with the respective sub-sample size, one can see that the number of diagnoses are quite equally distributed across males (mean=2.71, std=1.62) and females (mean= 2.71, std= 1.55). We thus deem it unlikely that descriptive differences between males and females (e.g. subtle variance differences) might have driven the association results. We explore other potential explanations below.

Supplementary Fig. 6: Distribution of psychopathology measures in the HBN sample as total counts and percentages stratified for sex. Diagnoses were grouped into six distinct categories following ³. ADHD= Attention Deficit Hyperactivity Disorder, ASD= Autism Spectrum Disorder, ND= Neurodevelopmental Disorder

One potential point of explanation might be the diagnosis itself. Figure C and D illustrate the distribution of individual primary diagnoses in our sample for males and females. Please note that the diagnoses were grouped into the six categories, i.e., ADHD, Autism Spectrum Disorder, Mood Disorder, Anxiety Disorder, Other Disorders, and other neurodevelopmental disorders, following an approach from Voldsbekk et al. (2023; DOI: 10.1101/2023.03.31.23288009). One can see that females show higher rates of Anxiety and Mood related disorders, while male participants are more often diagnosed with ADHD or autism spectrum disorder.

One might now speculate that male-typical vs female-typical disorders may factor into the explanations for female-specific associations with our brain embedding scores. Specifically,

male-typical disorders typically emerge earlier in life (see GBD 2019 Mental Disorders Collaborators; DOI: 10.1016/S2215-0366(21)00395-3) and our puberty sensitive brain embedding score may therefore not be related to those disorders in the same degree as the more puberty-sensitive disorders such as anxiety and mood which have a higher prevalence in females. This might be supported by the observation that the puberty – brain embedding associations in the HBN also pointed towards stronger associations in females, though important to note that these were not significant following correction. Since we cannot directly test the putative hypothesis that male-prevalent vs female-prevalent disorders can directly explain the observed association pattern - due to sample sizes restrictions in the clinical categories - further research is needed to determine the exact cause of this pattern. We have now incorporated this additional information and putative explanation in the manuscript (p14. 383ff):

“In females only, we observed small yet significant effects of psychopathology severity on the brain embedding score. By accounting these analyses for age, we ruled out that these associations simply mimic a larger number of diagnoses with increasing age. Since descriptive statistics of the number of diagnoses were almost identical across sexes (Supplementary Fig. 6), we deem it unlikely that such subtle variances might have driven differences in association strength. A more likely explanation might be the diagnoses themselves, as we observed diagnosis distributions matching the known patterns of more male-prevalent (e.g., ADHD) vs. more female-prevalent (e.g., mood or anxiety) disorders. Thus, it may be possible that the derived brain embeddings are more sensitive to female-prevalent disorders. Even though puberty associations did not reach statistical significance in the HBN sample, they pointed towards higher effects in females. As many female-typical mental disorders emerge during puberty⁴² our brain embedding may also have a higher sensitivity towards those disorders. However, since we were not able to test this directly because of sample size restrictions, this line of reasoning should be considered as hypothesis generating and needs to be investigated in future research.”

42. GBD 2019 Mental Disorders Collaborators. Global, regional, and national burden of 12 mental disorders in 204 countries and territories, 1990–2019: a systematic analysis for the Global Burden of Disease Study 2019. *Lancet Psychiatry* 9, 137–150 (2022).

8. Authors mention they only used the first embedding as it captures the highest variance. However, given that they are computing associations with puberty and psychopathology, both of which are likely to exhibit a decent amount of variability at the individual level, can the authors comment on how integrating additional embeddings in future analyses may affect the results?

We agree with the reviewer that integrating additional brain embeddings might indeed reveal interesting associations with puberty and/or psychopathology beyond the results we have described in the manuscript. Our work can be seen a first proof of concept of the pipeline for learning brain embeddings via domain adaptation. We therefore decided to restrict our investigations to the first brain embedding which captures the most variance of between-subject similarities, akin to the general idea of a Principal Component Analysis. It remains to be investigated to what degree subsequent brain embeddings contain additional information associated with psychopathology and puberty.

Diffusion map embedding on networks derived via similarity network fusion is still an underexplored research approach. To the best of our knowledge, only Markello et al. (2021; DOI: 10.1038/s41531-020-00144-9; cf Figure 4) have started to investigate what kind of information is encoded in (later) brain embeddings. Their interesting results indicate that later embeddings can capture both uni- or multimodal information. This supports that there may be relevant information to explore in these brain embeddings, yet also highlights the need for further investigations such as how noise is represented among them, how reproducible they are, and which modalities together shape multimodal vs. unimodal representations of the data. We believe this groundwork is necessary before testing for associations with psychopathology and puberty and have therefore decided to focus on the first brain embedding now, while targeting other brain embeddings in a dedicated study.

We added a brief discussion on the matter to the manuscript (p.16, l. 435ff.):

“However, beyond the proof-of-concept of the current study, we nevertheless deem it important to extend our approach with additional (imaging) modalities to tests its generalizability beyond the two imaging features. Furthermore, integrating additive data sources may result in a more holistic (i.e., multimodal) phenotype representing brain development or dynamics which may help to explain additional variance in behavioral or mental health measures and thus may substantiate the brain embedding score utility in capturing brain trajectories. In addition, focusing on more than the first brain embedding might also help to explain additional variance in the tested associations. However, we consider it essential for future work to systematically test how modality-specific information is encoded in the brain embeddings before testing if later embeddings contain biologically meaningful between or within subject variances.”

9. Was psychopathology considered in ABCD at all? Were subjects with diagnoses excluded from the analyses? Did authors consider using participants with known diagnoses in ABCD to validate their results regarding the psychopathology associations obtained in HBN?

We initially did not include any measures of psychopathology in the ABCD study. Our initial approach was to use ABCD with its longitudinal design for puberty associations and to test for clinical associations in a dedicated clinical cohort – the HBN.

However, we agree that clinical patterns may also be tested with dimensional psychopathology measures directly in ABCD. We now included analyses aiming at replicating our association analyses across cohorts: For the ABCD sample this means that we now also include association analyses related to psychopathology, and for HBN we include associations with puberty. Regarding psychopathology in the ABCD sample, we decided to stick to a dimensional measure of psychopathology, i.e., the CBCL total score. Since the ABCD study constitutes a population sample, there is simply not enough variance in measures like the number of diagnosis. In particular, relying on ‘present’ or ‘current’ diagnoses from the KSAD inventory based on the caregiver reports (see Barch et al. 2021; DOI:

10.1016/j.dcn.2021.101031; Table S2 which shows that more modules of the KSAD are administered to caregivers compared to youth) our N=7776 participants only had an average of 0.33 (0.77 std) diagnoses. Using the CBCL total score thus allowed us to investigate psychopathology on a population level, with scores that might not (yet) exceed any clinical rating threshold.

Linear models were performed in similar vein to the association analysis with the puberty variables: we calculated two cross-sectional models with age and site as covariate and one longitudinal model in which the delta measures of the brain embedding, the change CBCL score and the age difference between both timepoints were integrated.

Importantly, we observed associations between the total CBCL score and the predicted brain embedding at both timepoints, adding support to the findings previously obtained in the (clinical) HBN sample (baseline_{female}: $b = -.005$, $p = 2.86 \times 10^{-5}$, $h^2 = .005$, $N = 3586$; baseline_{male}: $b = -.007$, $p = 8.42 \times 10^{-13}$, $h^2 = 0.01$, $N = 4189$; follow-up_{female}: $b = -.004$, $p = .002$, $h^2 = .004$, $N = 3099$; follow-up_{male}: $b = -.006$, $p = 2.24 \times 10^{-6}$, $h^2 = .006$, $N = 3602$). Of note, the CBCL information is only available based on caregiver reports. For the change score associations we did observe a significant effect for males ($b = -.001$, $p = .005$, $h^2 = .002$, $N = 3602$), while for females the association was not significant (corrected alpha $.05/6 = .0083$; $b = -.0002$, $p = .50$, $h^2 = .0002$, $N = 3098$). See also the Figure below:

Supplementary Fig. 4: Associations between brain embeddings and dimensional psychopathology, both in cross-sectional and longitudinal data of the ABCD cohort. First two columns refer to associations between predicted brain embeddings and the respective psychopathology (CBCL Total) per timepoint. Δ refers to the association between the Δ brain embedding and the Δ CBCL score. Annotations refer to effect sizes from linear models and hashed cells indicate non-significant results. Linear models were performed in similar vein to the puberty analyses in the main manuscript: cross-sectional models were calculated with age and site as covariate and the longitudinal model was complemented with site and Δ age between both timepoints. Exact p-values and effect sizes can be found in Supplementary Table 1.

While psychopathology effects in HBN were already rather small, as expected, associations in the ABCD were even smaller - yet significant. Of note, in all models besides female baseline, age effects were descriptively slightly larger compared to the psychopathology effects. For the change models, one must note that the distributions for the CBCL total score followed a narrow

gaussian with most participants having only minor changes centered around zero. We now include this new information in the manuscript's results section (p.11, l. 287ff.) and refer to the Supplementary Material which now also incorporates all relevant information on these analyses:

“Since dimensional measures of psychopathology allow to test these associations also in a non-clinical sample, we aimed at validating these findings in the cross-sectional and longitudinal data from the ABCD sample. Here, we observed significant effects (all $p < .005$) between the CBCL score and the brain embedding, for all cross-sectional models (males and female) and the longitudinal model in males (Supplementary Fig. 4 and Supplementary Table 1 for details and exact effect sizes).”

Based on the small effects one should be cautious with too extensive interpretations, however we consider it important to highlight that our model seems to be also sensitive to subtle psychopathological manifestations in a non-clinical, population sample. We now do so in the discussion section of our manuscript (p. 15, l. 418ff.):

“To overcome sample size restrictions often observed in clinical samples, leveraging the longitudinal data from ABCD may further help to investigate the marker's sensitivity to capture refined, but biologically meaningful, mental health processes related to brain dynamics⁴⁶. Our validation analyses in the ABCD sample yielded small – yet significant – effect for the psychopathology associations, indicating that our model may also be sensitive to subtle psychopathological manifestations that do not (yet) exceed a clinical threshold. These initial results suggest that our approach may also render useful to study psychopathology in future releases of the ABCD sample.”

46. Solmi, M. *et al.* Age at onset of mental disorders worldwide: large-scale meta-analysis of 192 epidemiological studies. *Mol. Psychiatry* 27, 281–295 (2022).

10. Please reference Figure 1B in the text and provide some context for it.

We have now added a reference for Figure 1B in the main text (p.13, l. 336):

“Of note, all analyses were performed stratified for sex, because the brain embeddings span a sex-gradient (see Figure 1B), [...]”

Reviewer #2

A great big data project. The authors present a relatively new multi-dimensional data reductions technique and then used a supervised domain adaption to make it transferable to other datasets. The study is commendable on its approach, and provides a valuable contribution. However, I have a few suggestions and concerns that should be addressed to improve the manuscript.

We would like to thank the reviewer for the positive feedback and fruitful comments.

1. Firstly, I acknowledge that it is a very technical approach, but throughout the whole paper there is room for improvement in the writing being more accessible for a general audience.

We apologize if some technical details were not clear. We have carefully read through the manuscript and have tried improving clarity to a general audience. We have also added several new figures and tables as to provide the reader a better insight into the underlying data. Furthermore, in response to comment #3 below, we have updated the introduction with additional literature on puberty and psychopathology, which has reduced the technical focus of the introduction.

2. I'm not convinced of the use of the term "signatures of pubertal brain development" used in the title. The study simply found an embedding from multidimensional data that captured some variance in puberty that differed in only females in only late puberty. That is not to say the finding is not worthwhile but does not quite capture the comprehensive and specific characterisation typically associated with a 'signature'.

We agree with the reviewer that 'signature' may not be the best term and have changed to 'traces' throughout. This has also led to a change in the title, which now reads "Traces of pubertal brain development and health revealed through domain adapted brain network fusion".

3. The introduction content is just about the method, and does not raise any literature around what is known about brain development associated with puberty or with psychopathology.

We thank the reviewer for pointing this out. We have now extended our Introduction with a section dedicated to brain development associated with puberty and its link to psychopathology (p. 3, l. 97ff):

"We validated our approach in an unseen longitudinal sample from the Adolescent Brain Cognitive Development (ABCD) Study¹⁴ and on a clinical population of subjects from the Healthy Brain Network (HBN)¹⁵ sample. Both datasets allow to investigate unique processes shaping the human brain in development, specifically pubertal maturation, and emerging psychopathology.

Puberty depicts a phase of biological and psychological changes potentially mediated by neurodevelopment beyond the effect of age^{16–18}. Variables assessing pubertal status can thus be more sensitive measures than age for studying brain maturation in youth (e.g.,¹⁹). Previous work revealed global reductions in cortical grey matter volumes and thickness with advanced pubertal maturation, with evidence from both, cross-sectional and longitudinal data. These effects appear to be distributed across the whole cortex rather than being circumscribed to a specific set of regions (see¹⁷ for a review). However, as different studies use different approaches to account for age and sex, inconsistencies exist in terms of effect sizes and effect directions, including those of opposing effect directions in males and females^{20,21}. These conflicting observations might arise from certain methodological choices but also from individual variability in pubertal timing and progression through maturational stages. While all adolescents undergo the same pubertal stages, there is quite some variability regarding pubertal onset and tempo of changes, which has been linked to mental health conditions^{22–24}. In females, earlier pubertal timing appears to be associated to worse mental health conditions (e.g.,^{22,25}), while for boys both very early and very late onset has been linked to worse psychological outcome (e.g.,^{26,27}).

Given the close interplay between pubertal maturation, brain development and its link to emerging psychopathology, we aimed at investigating the sensitivity of brain embeddings toward these two entities. We show that our model can reveal traces of pubertal brain development and allows to capture biological variance related to emerging psychopathology, suggesting its utility in investigating within-person changes in youths.”

14. Casey, B. J. *et al.* The Adolescent Brain Cognitive Development (ABCD) study: Imaging acquisition across 21 sites. *Dev. Cogn. Neurosci.* 32, 43–54 (2018).
15. Alexander, L. M. *et al.* An open resource for transdiagnostic research in pediatric mental health and learning disorders. *Sci. Data* 4, 170181 (2017).
16. Blakemore, S.-J., Burnett, S. & Dahl, R. E. The role of puberty in the developing adolescent brain. *Hum. Brain Mapp.* 31, 926–933 (2010).
17. Vijayakumar, N., Op de Macks, Z., Shirtcliff, E. A. & Pfeifer, J. H. Puberty and the human brain: Insights into adolescent development. *Neurosci. Biobehav. Rev.* 92, 417–436 (2018).
18. Suleiman, A. B., Galván, A., Harden, K. P. & Dahl, R. E. Becoming a sexual being: The ‘elephant in the room’ of adolescent brain development. *Dev. Cogn. Neurosci.* 25, 209–220 (2017).
19. Holm, M. C. *et al.* Linking brain maturation and puberty during early adolescence using longitudinal brain age prediction in the ABCD cohort. <http://medrxiv.org/lookup/doi/10.1101/2022.05.16.22275146> (2022) doi:10.1101/2022.05.16.22275146.
20. Beck, D. *et al.* Puberty differentially predicts brain maturation in males and females during early adolescence: A longitudinal ABCD Study. <http://medrxiv.org/lookup/doi/10.1101/2022.12.22.22283852> (2022) doi:10.1101/2022.12.22.22283852.
21. Wierenga, L. M. *et al.* A Key Characteristic of Sex Differences in the Developing Brain: Greater Variability in Brain Structure of Boys than Girls. *Cereb. Cortex N. Y. N* 1991 28, 2741–2751 (2018).
22. Dehestani, N., Whittle, S., Vijayakumar, N. & Silk, T. J. Developmental brain changes during puberty and associations with mental health problems. *Dev. Cogn. Neurosci.* 60, 101227 (2023).
23. Kaczurkin, A. N., Raznahan, A. & Satterthwaite, T. D. Sex differences in the developing brain: insights from multimodal neuroimaging. *Neuropsychopharmacology* 44, 71–85 (2019).
24. Pfeifer, J. H. & Allen, N. B. Puberty Initiates Cascading Relationships Between Neurodevelopmental, Social, and Internalizing Processes Across Adolescence. *Biol. Psychiatry* 89, 99–108 (2021).
25. Copeland, W. E., Worthman, C., Shanahan, L., Costello, E. J. & Angold, A. Early Pubertal Timing and Testosterone Associated With Higher Levels of Adolescent Depression in Girls. *J. Am. Acad. Child Adolesc. Psychiatry* 58, 1197–1206 (2019).
26. Mendle, J. & Ferrero, J. Detrimental psychological outcomes associated with pubertal timing in adolescent boys. *Dev. Rev.* 32, 49–66 (2012).
27. Conley, C. S. & Rudolph, K. D. The emerging sex difference in adolescent depression: Interacting contributions of puberty and peer stress. *Dev. Psychopathol.* 21, 593–620 (2009).

4. It is great the models were tested in independent datasets. The study used three cohorts (PNC, ABCD, HBN): PNC for training of 'typical brains', ABCD for examining puberty and HBN for examining psychopathology. However, each of the three cohorts used, all have measures of puberty and psychopathology. I don't understand why there was not an attempt to replicate across cohorts.

We agree with the reviewer's comment and have in the revised version added new analyses.

First, we would like to clarify that the PNC sample was solely used in the model building step, that is, our model did not yield any predictions for the PNC sample. Accordingly, we did not and cannot use the PNC sample for any association analyses. We would also briefly like to clarify the rationale for our initial focus on puberty in ABCD and psychopathology in HBN. The longitudinal design of the ABCD study allows us to map the process of pubertal development within subjects longitudinally, which is why we chose to select ABCD for investigations into pubertal development. Likewise, the HBN is a sample with mostly clinically diagnosed individuals, which is why it was well suited for investigations into psychopathology.

(1) We have previously already included data on puberty in the analysis of the HBN data because of the close relationship between pubertal maturation and potentially emerging disorders during that period. However, as reported in the manuscript, we did not find any association for PDS in those analyses, which may relate to the broader age range and distributions in pubertal categories. We however agree that the previous analyses did not constitute a direct replication attempt of the ABCD puberty associations given the differences in age range, and we therefore added a new analysis on puberty associations in a subsample of the HBN that matched the age range of ABCD. Specifically, since ABCD was assessed at two time points with different age distributions, we performed two sub-analyses, one subsampling the HBN to the age range of the ABCD baseline data ($n=141$ females and $n=300$ males; aged 8.92 – 11.08 years), and one subsampling HBN to the age range of the ABCD 2-year follow up data ($n=134$ females, $n=259$ males; aged 10.59 to 13.83 years). In each of the two HBN subsamples, we then performed the same association analyses as performed in ABCD (i.e., 'embedding ~ PDS_mean + age + site'). We did not observe statistically significant associations, neither in the first subsamples (males: $b = .07$, $p = .07$, $h^2 = .003$; females: $b = .33$, $p = .07$, $h^2 = .02$), nor in the second subsample (males: $b = -.12$, $p = .40$, $h^2 = .01$; females: $b = -.01$, $p = .95$, $h^2 = .002$).

The lack of significant puberty associations in the HBN subsamples might have been driven by different factors: Firstly, the HBN constitutes a clinical sample, and clinical profiles may interfere with pubertal development, which hampers a 1:1 replication. Secondly, although significant, effect sizes in ABCD were only small to moderate in the ABCD sample, a certain sample size is needed to detect such effects and thus it is not surprising that with much fewer subjects (only 140 females following age matching), the effects did not pass statistical significance thresholding.

We now include this information in the Result section of our manuscript: (p. 6. l. 197ff.) and refer to the Supplementary Material which now also incorporates all relevant information on these analyses:

“Aiming at replicating these puberty associations in the clinical HBN sample, we performed two additional analyses in which we subsampled the HBN sample to the age of the ABCD baseline and the ABCD follow-up data. Calculating the same cross-sectional puberty models in these HBN subsets did not yield statistically significant results (Supplementary Table 2).”

And further discuss it in our Discussion (p. 14, l. 399ff.):

“[...] When extending the model with pubertal variables, we did not observe additional effects, which may be related to prevalence of emerging psychopathology in the HBN sample, which may interfere with puberty and may thus explain, why we were not able to replicate the cross-sectional puberty associations in the HBN sample. Furthermore, subsampling the HBN sample to the respective age ranges of the ABCD visits decreased the available sample size in a way that has not sufficient statistical power to detect small to moderate effects.”

(2) We previously targeted psychopathology in a clinical sample because ABCD is a relatively healthy sample (at this point in time) and may therefore not be the most sensitive to study such associations. When looking at a similar measure of psychopathology as the one used in HBN (number of diagnoses), the average was only 0.3 diagnoses per subjects, suggesting that the sample may be too healthy to directly attempt at replicating the effect observed in HBN. However, we agree with the reviewer that the availability of psychopathology data in both samples is a strong opportunity for replication analyses. We therefore now added a new analysis in which we included associations with dimensional estimates of psychopathology, in both the HBN and the ABCD sample. Since the reviewer raised a related point with regard to the use of psychopathology data in HBN, we here only show the ABCD specific results, and show the HBN results in response to comment #9 below.

In the ABCD sample, linear models were performed in similar vein to the association analysis with puberty variables: Using the CBCL score as a measure of psychopathology, we tested for associations with the brain embeddings while controlling for age and site. Specifically, we calculated three models, one with baseline data, one with 2-year follow up data, and one longitudinal with deltas of the brain embedding. Of note, the CBCL information is only available based on caregiver reports. We observed significant associations between the total CBCL score and the predicted embedding at both timepoints ($baseline_{female}$: $b = -.005$, $p = 2.86 \times 10^{-5}$, $h^2 = .005$, $N = 3586$; $baseline_{male}$: $b = -.007$, $p = 8.42 \times 10^{-13}$, $h^2 = 0.01$, $N = 4189$; $follow-up_{female}$: $b = -.004$, $p = .002$, $h^2 = .004$, $N = 3099$; $follow-up_{male}$: $b = -.006$, $p = 2.24 \times 10^{-6}$, $h^2 = .006$, $N = 3602$). For the change score associations, we did observe a significant effect for males ($b = -.001$, $p = .005$, $h^2 = .002$, $N = 3602$), while for females there was no significant association ($b = -.0002$, $p = .50$, $h^2 = .0002$, $N = 3098$). See also the Figure below:

Supplementary Fig. 4: Associations between brain embeddings and dimensional psychopathology, both in cross-sectional and longitudinal data of the ABCD cohort. First two columns refer to associations between predicted embeddings and the respective psychopathology (CBCL Total) per timepoint. Δ refers to the association between the Δ embedding and the Δ CBCL score. Annotations refer to effect sizes from linear models and hashed cells indicate non-significant results. Linear models were performed in similar vein to the puberty analyses in the main manuscript: cross-sectional models were calculated with age and site as covariate and the longitudinal model was complemented with site and Δ age between both timepoints. Exact p-values and effect sizes can be found in Supplementary Table 1.

While psychopathology effects in HBN were already rather small, as expected, associations in the ABCD were even smaller - yet significant. Of note, in all models besides female baseline, age effects were descriptively slightly larger compared to the psychopathology effects. For the change models, one has to note that the distributions for the CBCL total score followed a narrow gaussian with most participants having only minor changes centered around zero.

We now include this new information in the manuscript's results section (p. 11, l. 287ff.) and refer to the Supplementary Material which now also incorporates all relevant information on these analyses:

“Since dimensional measures of psychopathology allow to test these associations also in a non-clinical sample, we aimed at validating these findings in the cross-sectional and longitudinal data from the ABCD sample. Here, we observed significant effects (all $p < .005$) between the CBCL score and the brain embedding, for all cross-sectional models (males and female) and the longitudinal model in males (Supplementary Fig. 4 and Supplementary Table 1 for details and exact effect sizes).”

Since coefficients were rather small one should be cautious with too extensive interpretations, however we consider it important to highlight that our model seems to be also sensitive to subtle psychopathological manifestations in a non-clinical, population sample. We therefore include the following in the discussion section (p. 15, l. 422ff.):

“Our validation analyses in the ABCD sample yielded small – yet significant – effect for the psychopathology associations, indicating that our model may also be sensitive to subtle psychopathological manifestations that do not (yet) exceed a clinical threshold. These initial

results suggest that our approach may also render useful to study psychopathology in future releases of the ABCD sample.”

5. I would be interested to hear why the authors chose to use surface area and volume, rather than surface area and cortical thickness (which are more typically used) or even all three. Volume is just a function of surface area and thickness, so will be highly colinear with surface area.

The reviewer raises a valid point about the collinearity between the incorporated modalities. Indeed, we here purposefully targeted the known relationship between area and volume, aiming to exploit the collinearity between these measures to derive a robust and non-sparse reference space. Our work can be seen as a proof of concept of a framework that can naturally be extrapolated to many other modalities, including cortical thickness, but also other measures such as DTI or fMRI. We here naturally fused two modalities of which we expected good fusion ability, in particular as it has been shown that features from different modalities can have a high degree of longitudinal inter- and intraindividual variability (Bottenhorn et al. 2022; DOI: 10.1101/2022.12.19.521089). To deploy our framework, we choose brain volume and area because both follow a comparable (normative) developmental trajectory from late childhood into adolescence (Bethlehem et al., 2022; DOI: 10.1038/s41586-022-04554-y), allowing us to mitigate imaging modality-induced variances when testing the sensitivity of the resulting brain embedding to processes shaping brain development. Nonetheless, we agree with the reviewer that the integration of other modalities resembles a viable research target. In our previous manuscript version, we had already acknowledged that our focus on volume and area depicts a limitation of our current work and that we deem it important to extend our approach with additional modalities to test its generalizability in future work. Related to integrating additional or different imaging modalities, another worthwhile consideration for future work may be the use of additional brain embeddings (see our Response to Reviewer #1, comment #6).

6. The training dataset has quite a large age range. Did you examine to what extent the first embedding is related to age (and overall brain size)? Was any attempt made to exclude psychopathology from the training dataset in PNC?

The large age range of the PNC data was intended, as we were particularly interested in building a modelling framework that is applicable to samples with unique characteristics, such as the longitudinal design from childhood into young adulthood as targeted by the ABCD. Thus, building our model upon the PNC data will allow us to apply the same model to upcoming releases of the ABCD study, without the need to train in other source datasets. We use PNC as a population sample for modelling and did not attempt at excluding participants based on psychopathology or any other variable of interest in this sample.

The first brain embedding did not correlate with age neither in the first, nor the second timepoint (both $r \sim -0.02$, $p \sim 0.12$) in the ABCD data. In the HBN sample, age correlated significantly with the brain embedding score ($r = -0.09$, $p = 8.32 \times 10^{-6}$). Overall brain size – operationalized as total intracranial volume – did significantly correlate with the brain embeddings across both timepoints in the ABCD (both $r \sim .80$, $p = 0.0$; p -value lower than machine precision) and the HBN sample ($r = -.60$, $p = 5.62 \times 10^{-217}$), which is not surprising,

given that the derived brain embeddings show a strong sex gradient (see Figure 1B in main manuscript), which separates males and females. Of note, the negative correlation in the HBN refers to the fact that the sex gradient is flipped due to the way the embedding is calculated.

7. Would it be possible to visualise the brain features (regional volume and surface area) associated with the first embedding? And that captured the most variance between individuals and over age (in the normative model), and associated with both puberty and psychopathology?

Since our brain embedding is associated with puberty and psychopathology and allowed us to model variance between individuals in a subject similarity reference space, we agree that it would be valuable to highlight the brain sources underlying the brain embeddings. We therefore now mapped the embeddings back to raw feature space and now include two additional figures in the Supplementary Material (Supplementary Fig. 1) that show the correlation between raw features and the brain embedding. To derive these figures, we calculated pairwise correlations between the predicted brain embedding in the HBN and ABCD sample and each individual brain feature, both for area and volume measures, respectively. As one can see, brain maps revealed a comparable pattern across datasets, with fronto-parietal and temporal regions depicting highest correlations.

We now include this information also in the main manuscript (p. 6, l. 163) and (p.20, l. 576ff.):

“Brain maps illustrating the associations between brain embeddings and raw features showed similar patterns in both samples (Supplementary Fig. 1).”

“Additionally, Supplementary Fig. 1 depict the correlation between the raw features and the first brain embedding in the HBN and ABCD sample, respectively.”

Supplementary Fig. 1: Correlation between raw brain features and first brain embedding. Correlation values indicate Pearson correlation coefficients. Visualization was performed with *ggseg* (version 1.6.5¹) in R (version 4.2.3). Since algebraic signs of the embeddings might be positive or negative (i.e., akin to an eigenvector), we multiplied negative correlations by (-1) to derive at comparable patterns across cohorts. Spin permutation testing with 5000 iterations indicated significant correlations between brain maps of the ABCD and HBN dataset (all $r > 0.64$, all $p < 0.0004$). Permutations were performed using the ENIGMA Toolbox (version 2.0.3,²).

Supplementary References:

1. Mowinckel, A. M. & Vidal-Piñeiro, D. Visualization of Brain Statistics With R Packages *ggseg* and *ggseg3d*. *Adv. Methods Pract. Psychol. Sci.* 3, 466–483 (2020).
2. Larivière, S. *et al.* The ENIGMA Toolbox: multiscale neural contextualization of multisite neuroimaging datasets. *Nat. Methods* 18, 698–700 (2021).

8. When investigating puberty, there is established literature (even specifically in the ABCD cohort) around the need to covary for BMI, SES and race. This was not addressed or mentioned.

We thank the reviewer for pointing this out. Accordingly, we have now added BMI, SES and race / ethnicity as covariates to the linear models relating pubertal development and the brain embedding in the ABCD study.

Briefly, BMI was derived by dividing an individual's height (lb) by their height in inches (in) squared and multiplied by 703. Height and weight measures were averaged across two measurements for which the most data points were available. BMI values smaller than 10 or higher than 50 were excluded after manual inspection suggesting erroneous weight or height measurements. Race/ethnicity was directly derived from the ABCD data release (acspsw03) with 5 different levels indicating: white, black, hispanic, asian, other. The latter referred to individuals identified as multiracial and/or belonging to a race/ethnicity with too few members in the sample. For SES, "refuse to answer" and "don't know" responses were encoded as missing (NaN). We then performed a rank-based inverse normal transformation (palm_inormal) on parental education and total family income to derive normally distributed variables for which we consequently calculated the average representing an overall SES status.

Linear models were calculated in a similar vein to the initial association analysis described in the manuscript: we ran two cross-sectional models with age, SES, BMI, and race/ethnicity as covariates. Of note for SES there was only baseline data available, thus limiting the covariate to the baseline model. Additionally, we ran a longitudinal model in which the delta measures of the brain embedding, the change PDS score and the age and BMI difference between both timepoints were integrated. Moreover, baseline SES and race/ethnicity was added as covariate without a change score.

For the two cross-sectional association analyses at baseline and follow-up, covarying for SES, race/ethnicity, and BMI as covariates diminished the reported associations with puberty (all $p > .02$), both in males and in females (implications discussed below). In the longitudinal analyses, however, the covariates did not affect the identified associations with puberty.

Specifically, we observed significant associations of puberty with Δ brain embedding for females both when controlling (caregiver $b = -.07$, $p = 2.51 \times 10^{-9}$, $h^2 = .01$, $N = 2636$; youth: $b = -.08$, $p = 4.02 \times 10^{-8}$, $h^2 = .03$, $N = 1107$) or not controlling for SES, BMI and race/ethnicity (caregiver: $b = -.06$, $p = 2.37 \times 10^{-10}$, $r^2 = .02$, $N = 3135$; youth: $b = -.08$, $p = 3.79 \times 10^{-11}$, $r^2 = .04$, $N = 1375$). Puberty associations in the male model were already not significant without controlling for these confounds, which did not change when adding them in. See also the figure below:

Supplementary Fig. 2: Associations between brain embeddings and puberty, both in cross-sectional and longitudinal data of the ABCD cohort. First two columns in A) and B) refer to associations between predicted brain embeddings and the respective pubertal score (PDS mean) per timepoint. Δ refers to the association between the Δ brain embedding and the Δ PDS mean score. Annotations refer to effect sizes from the linear models and highlighted cells with dashed lines indicate significant results. Linear models were performed in similar vein to the puberty models in the main manuscript but were expanded with Body Mass Index (BMI), Socioeconomic Status (SES), and race/ethnicity as covariates: we ran two cross-sectional models with age, SES, BMI, and race/ethnicity as covariates. Of note for SES there was only baseline data available. Additionally, we ran a longitudinal model in which the delta measures of the brain embedding, the change PDS score and the age and BMI difference between both timepoints were integrated. Moreover, baseline SES and race/ethnicity was added as covariate without a change score. $BMI = (\text{height (lb)} / \text{height (in)}) \times 703$. Height and weight measures were averaged across two measurements. BMI smaller than 10 or higher than 50 were excluded after manual inspection. Race/ethnicity (ABCD variable acspsw03) refers to 5 different levels: White, Black, Hispanic, Asian, other. Other includes individuals identifying as multiracial and/or belonging to a race/ethnicity with too few members in the sample. For SES variables (i.e., parental education and total family income), “refuse to answer” and “don’t know” were encoded as “NaN”. SES variables were transformed via rank-based inverse normal transformation and averaged resulting in a single SES status variable. Exact p-values and effect sizes can be found in Supplementary Table 1.

We now refer to these analyses in the Results section of our manuscript, where we also point to the supplementary figure. The legend of the figure provides additional details of the accompanying methods of the analysis (p 8, l. 233ff.):

“After accounting for Body Mass Index (BMI), socioeconomic status (SES), and race/ethnicity in the puberty association models, associations between Δ PDS and Δ brain embedding remained significant whereas cross-sectional associations did not (Supplementary Fig. 2 for methodological details and Supplementary Table 1 for exact model outcomes), further supporting sensitivity of the approach to longitudinal contexts.”

The fact that control for BMI, SES and race/ethnicity diminished associations in the cross-sectional but not in the longitudinal framework has several implications. First, it is worth noting that all analyses involving these covariates sometimes yielded smaller samples due to missing values in the covariates, which may affect power to detect effects. Second, the modelled confound variables are known to be correlated with the phenotypes of interest. There is substantial evidence suggesting that race/ethnicity, SES, and BMI are all closely related to pubertal timing and duration (see e.g., Cheng et al. 2021; DOI: 10.3389/fendo.2021.608575, Oelkers et al., 2020; DOI: 10.1159/000513787). In such scenarios it can be difficult to associate variance to distinct factors, however, if the identified association between brain embedding and puberty would only reflect confound effects, we would expect the longitudinal effects to diminish as well. Instead, longitudinal models remained significant even after controlling for BMI, SES and race/ethnicity. On a related note, since baseline puberty status (i.e., the intercept) may affect the slope, we repeated the longitudinal model with baseline PDS as an additional covariate among baseline BMI, SES, and race/ethnicity, and found the same effects. Of note, in both models for caregiver and youth, baseline BMI was also associated with the brain embedding in females, yet to a lower extent than puberty. We would therefore argue that our new analyses and in particular the longitudinal models capture variance beyond the covariates and thus might indeed help to resolve the ambiguity of cross-sectional approaches. In future studies one could try to alleviate covariance with these variables already in the training phase of the machine learning model, for example by pre-residualizing brain features for BMI, SES and race/ethnicity, or by domain adaptation with regard to specific groups such as high vs low BMI. Our domain adaptation framework may constitute a perfect setting for handling such covariates as it allows to build models targeted at specific subgroups, in which one could then investigate the pre-or absence of specific effects.

We now include these information and discussion in the manuscript: (p.13, l. 342ff.)

“After adding BMI, SES, and race/ethnicity^{37,38} as covariates into our model, cross-sectional effects diminished. This aligns with reports suggesting a close link between those factors and pubertal timing and duration (e.g.,^{38,39}). Given the high inter-correlation between the studied variables, it may be difficult to disentangle variance to distinct components. Therefore, we argue, that longitudinal analyses may help to resolve the ambiguity of the cross-sectional analyses. Since the ABCD study offers an unprecedented resource for granular investigations of child and adolescent brain and pubertal maturation, we leveraged the longitudinal data of the ABCD cohort and investigated whether the Δ brain embedding, that is the difference between the two predicted brain embedding scores for baseline and the 2-years follow up data, can serve as an additional marker for brain trajectories. Pubertal associations with the Δ brain embeddings were significant for females, but not for males, which appeared to align with the pubertal maturation in females in the studied time period. The same pattern was observable when controlling for BMI, SES, and race/ethnicity, supporting that the reported cross-sectional puberty effects do not simply represent differences in these confounding factors either.”

37. Herting, M. M. *et al.* Correspondence Between Perceived Pubertal Development and Hormone Levels in 9-10 Year-Olds From the Adolescent Brain Cognitive Development Study. *Front. Endocrinol.* 11, 549928 (2021).
38. Cheng, T. W. *et al.* A Researcher’s Guide to the Measurement and Modeling of Puberty in the ABCD Study® at Baseline. *Front. Endocrinol.* 12, 608575 (2021).
39. Oelkers, L. *et al.* Socioeconomic Status Is Related to Pubertal Development in a German Cohort. *Horm. Res. Paediatr.* 93, 548–557 (2020).

9. I am not convinced that the approach taken to examine psychopathology is the most appropriate or interpretable. The authors use the total number of comorbid diagnoses as the marker of severity. This range is 1-10 comorbid diagnoses (mean 2.1, sd 1.6), so very unlikely individuals have more than 3-4 diagnoses. It also suggests for example, an individual with severe schizophrenia is not as impacted and an individual that only just meets criteria of ADHD and depression). Whilst the authors do acknowledge that it is a coarse measure of psychopathology, even without looking at different diagnoses separately, the cohorts have broad spectrum dimensional measures (such as the CBCL), that could not only be used across the entire cohort (typical and disorders), but would also provide greater variance in the psychopathology measure. At the very least a histogram should be provided with a) the number of different comorbidities (your unit of psychopathology severity), and b) one showing the prevalence of each disorder in the sample.

These are valid remarks and we have therefore incorporated several new analyses and a new figure to address the points.

Relating to the reviewers' suggestion to add a visualization that helps to better disentangle how our initial measure of severity of psychopathology and diagnosis are distributed across our HBN sample, we have now added Supplementary Figure 6 which one can find below this paragraph. In the upper row of the figure, we show how the sum of diagnosis is distributed across males and females (Panel A), which leads to the impression that males have much more diagnoses compared to females. However, weighting the overall count with the respective sub-sample size of males and females, respectively, one can see that the number of diagnoses is quite equally distributed across males and females. This is also mirrored when stratifying the mean and standard deviation across sex: female: mean= 2.71, std= 1.55, male: mean= 2.71, std=1.62). As suggested by the reviewer, we now also include information about diagnostic categories in the lower row of the figure: Panel C and D shows the distribution of individual primary diagnoses in our HBN sample. Please note that the diagnoses were grouped into the six categories, i.e., ADHD, Autism Spectrum Disorder, Mood Disorder, Anxiety Disorder, Other Disorders, and other neurodevelopmental disorders, following an approach from Voldsbekk et al. (2023; DOI: 10.1101/2023.03.31.23288009). The pattern that we observe matches previous reports (e.g., GBD 2019 Mental Disorders Collaborators) describing male and female prevalent disorders, such as females are more often diagnosed with anxiety related disorders in our samples, while males are more often diagnosed with ADHD.

Figure S6: Distribution of psychopathology measures in the HBN sample as total counts and percentages stratified for sex. Diagnoses were grouped into six distinct categories following ³. ADHD= Attention Deficit Hyperactivity Disorder, ASD= Autism Spectrum Disorder, ND= Neurodevelopmental Disorder

Supplementary References:

3. Voldsbekk, I. *et al. Delineating disorder-general and disorder-specific dimensions of psychopathology from functional brain networks in a developmental clinical sample.* <http://medrxiv.org/lookup/doi/10.1101/2023.03.31.23288009> (2023) doi:10.1101/2023.03.31.23288009.

Relating to the reviewer’s comment on dimensional measures of psychopathology, we would firstly like to reiterate that we consider our association with our proposed psychopathology measure as a showcase for our model’s applicability in samples with unique characteristics. Additionally, we like to emphasize that our hold-out HBN sample for which we report the association results do only contain participants with at least one diagnosis because we have used the healthy participants for domain adaptation. Therefore, we cannot investigate psychopathology across a ‘healthy - diseased’ spectrum as suggested by the reviewer. However, we agree that a dimensional measure may offer an additional test case with potentially more variance for our proposed analyses. Consequently, for the HBN sample, we repeated the association analyses reported in the manuscript but replaced the number of diagnoses with the CBCL_Total Score:

Using the CBCL Total score as an independent variable and including age and site as covariates, we did not observe a significant effect of psychopathology on the brain embedding for males ($b = .002$, $p = .06$, $h^2 = .004$, $N = 1269$), however, for females the effect reached statistical significance ($b = .004$, $p = .008$, $h^2 = .01$, $N = 635$). The identified association for psychopathology operationalized via the CBCL total score in females remained significant when controlling for puberty (PDS): $b = .005$, $p = .002$, $h^2 = .02$, $N = 471$.

Taken together, by using a dimensional measure for psychopathology, that is the CBCL total score, we did observe a pattern that almost perfectly resembles our initial results in which we used the sum of diagnosis as a proxy for psychopathology. As a side note, the number of diagnosis and the CBCL total score were correlated, i.e., $r_s = .31$, $p = 3.37 \times 10^{-46}$, which supports that the number of diagnoses may indeed serve as a proxy for severity in psychopathology.

In the manuscript, we have made the following changes to include the results described above (p. 11, l.279 ff.)

“Replacing the sum of diagnosis with a dimensional measure of psychopathology, i.e., the CBCL total score (Child-Behavior Checklist³⁴) we replicated the effects from the previous analysis. Specifically, we did not observe a significant effect of psychopathology on the brain embedding for males ($b = .002$, $p = .06$, $\eta^2 = .004$, $N = 1269$), however, for females the effect reached statistical significance ($b = .004$, $p = .008$, $\eta^2 = .01$, $N = 635$). The identified association for psychopathology operationalized via the CBCL total score in females remained significant when controlling for puberty (PDS): $b = .005$, $p = .002$, $\eta^2 = .02$, $N = 471$.”

Additionally, we also now have included the following changes in the discussion (p. 15, l.407ff.):

“We acknowledge that the sum of diagnoses in the HBN sample rather depicts a coarse measure of psychopathology, however, by expanding our work with a dimensional measure (the CBCL total psychopathology score), we did observe similar associations. Furthermore, both measures were moderately correlated ($r \sim .3$) supporting our initial approach to operationalize sum of diagnosis as a measure of psychopathology severity. Future research may leverage more fine-grained quantities, such as hierarchical representations of psychopathology (HiTOP⁴³) or different syndrome scales to better disentangle associations between the (Δ) brain embedding score and emerging mental health conditions during puberty. For example, dimensional approaches may further help to investigate whether the brain embedding scores are sensitive to capture neuronal traces of early pubertal timing (e.g., early menarche in females⁴⁴ and their relationship to internalizing psychopathology⁴⁵).”

43. Conway, C. C., Forbes, M. K. & South, S. C. A Hierarchical Taxonomy of Psychopathology (HiTOP) Primer for Mental Health Researchers. *Clin. Psychol. Sci.* 10, 236–258 (2022).

44. Mendle, J., Ryan, R. M. & McKone, K. M. P. Age at Menarche, Depression, and Antisocial Behavior in Adulthood. *Pediatrics* 141, e20171703 (2018).

45. Barendse, M. E. A. *et al.* Multimethod assessment of pubertal timing and associations with internalizing psychopathology in early adolescent girls. *J. Psychopathol. Clin. Sci.* 131, 14–25 (2022).

Minor

In the results it would be good to note what the change in PDS scores were for both males and females.

We have now added this information in the manuscript (p. 8, l. 213ff):

“Consequently, we repeated the linear models with Δ brain embedding as dependent and the Δ PDS scores (i.e., the puberty difference between baseline and 2-years follow up) as independent variable (caregiver report: female mean= 0.77, male mean= 0.38, youth report: female mean= 0.70, male mean= 0.21).”

We have also added a Supplementary Figure 5 which includes two violin plots showing the delta PDS score stratified for male and female and self-report responder. Please see our response to Reviewer 1, comment 5.

The sentence bottom page 5 is not clear and should be re-phrased. Also in that sentence, ‘adolescent’ should be ‘adolescence’.

We have revised the sentence as follows: (p. 7, l. 181ff.)

“We validated the biological utility of the predictions in capturing developmental brain dynamics by targeting puberty and mental health as two phenotypes that are closely related to each other. They both lay off their dynamics during adolescence and therefore are also intertwined with (developmental) brain trajectories^{24,29}.”

24. Pfeifer, J. H. & Allen, N. B. Puberty Initiates Cascading Relationships Between Neurodevelopmental, Social, and Internalizing Processes Across Adolescence. *Biol. Psychiatry* 89, 99–108 (2021).

29. Vijayakumar, N. *et al.* A longitudinal analysis of puberty-related cortical development. *NeuroImage* 228, 117684 (2021).

Reviewer #3

This is an interesting paper reporting on a novel approach in a proof of concept analysis. The approach combines SNF to derive similarity across subjects (embeddings) from an initial dataset (PNC), then uses machine learning model to train for prediction of brain embeddings based on two raw MRI features (cortical area, and volume) using both the original dataset and two subsets from unseen datasets (ABCD, HBN). The model was then validated on data from ABCD and HBN that was not used in the training phase to derive two predictions for ABCD longitudinal data (baseline, followup) and one prediction of brain embeddings for HBN data. Finally, two independent variables associated with brain development/health were tested to see if they predicted brain embeddings derived from SNF analysis: (i) pubertal development using longitudinal data from ABCD, and (ii) sum of diagnoses (a proxy of psychopathological severity) from HBN. The authors report that the model achieved high performance in unseen data, and that brain embeddings derived from unseen data were associated with meaningful variables (PDS and clinical diagnoses in females). The paper is well written, proof of potential utility is shown in the current paper and limitations of the data used and aspects of the approach are clearly laid out. I have the following questions/suggestions:

We would like to thank the reviewer for the positive feedback and fruitful comments.

1. what was the quality control approach used across datasets? were there key differences in data quality between ABCD and HBN datasets that may contribute to poorer model performance in HBN vs ABCD data? Did data quality in HBN also correlate with diagnostic sum (measure of psychopathology)?

Since we used PNC for modelling, our main emphasis was to ensure sufficient quality of the included data set in PNC on the one hand, while also maximizing sample size on the other. For this, we compared a machine learning model trained on all available PNC data to one trained in a subset that passed a quality control procedure based on Euler numbers (Rosen et al., 2017; DOI: 10.1016/j.neuroimage.2017.12.059). Specifically for the latter, we used FreeSurfer's Euler number as a proxy for data quality in the PNC sample by averaging left and right hemispheres measures into a single euler score. In analogy to Kaufmann et al (2019, DOI: 10.1038/s41593-019-0471-7) we then investigated which subjects exceeded three times the standard deviation below the mean euler number (note: more negative values refer to worse data quality) and excluded those (N=27) subjects from the PNC sample for model building. We then re-rerun our pipeline in the ABCD sample and compared the derived first brain embeddings for baseline and follow-up with the brain embeddings derived from the unrestricted sample initially reported in our manuscript. Prediction with the restricted PNC sample yielded almost identical model performance to the unrestricted sample reported in the manuscript. (baseline: $r=0.94$, $mse=0.914$, $rmse=0.956$, $mae=0.851$, $r^2=0.789$ || 2 years follow up: $r=0.94$, $mse=1.03$, $rmse=1.015$, $mae=0.913$, $r^2=0.785$). Accordingly, correlation analyses revealed almost identical brain embeddings indicated by significant correlation coefficient of about 0.999 for both timepoints. We therefore used the full sample to derive the final model as to maximize sample size for the machine learning. As noted by the reviewer, model performance was slightly lower in HBN than ABCD. We did not observe significant correlation between the number of diagnoses and data quality (euler number) in HBN ($r=0.03$, $p=0.11$). In our manuscript we have therefore suggested that the sample size used in the ABCD study for domain adaptation might have been the driving force behind better model performance, since the ABCD domain adaptation sample was approximately five times higher than the one

used in the HBN sample. Additionally, HBN consists of patients, and reduced model performance may therefore also to some degree reflect pathological variance on a neuronal level. Lastly, the narrow age range in the ABCD study may additionally support slightly better model performance.

2. I found the description of analyses examining associations bw brain embeddings and pubertal/clinical variables somewhat confusing at times. It would be helpful to use brain embeddings consistently when referring to embeddings to clarify for reader.

We appreciate the suggestion by the reviewer to enhance the clarity of the writing and have added 'brain' to the term 'embedding' throughout the manuscript.

3. It is notable that effect sizes for associations between brain embeddings and caregiver pubertal score are small-moderate for baseline data, small for 2 year data and very small for change scores (and results are weaker for youth report data). Given effect sizes are small, this method may require very large datasets for both training and testing which may limit applicability to clinical samples where large datasets with extensive characterization are limited.

We like to point out that our manuscript contains two distinct parts, i.e., (1) training a machine learning model with domain adaptation, and (2) model validation via association analyses. Both parts impose different sample size requirements.

(1) For our proposed machine learning method, a base data set is needed to train the relationship between brain data and brain embedding (here: the PNC sample). We show that high performance can be achieved with the ~1500 participants included. While this is indeed not a small sample, one can use openly available existing data sets for this base task. On top of the base data, the machine learning training requires a sample for domain adaptation, for which one can use a fraction of the target sample. As pointed out by the reviewer, clinical data might be limited, and may thus not be available for such an adaptation task. Our work demonstrates that we can use healthy participants from the target (e.g., clinical) sample for supervised domain adaptation, which in our opinion constitutes a strength of our proposed framework since "valuable" patient data in putative small samples do not necessarily need to be used in the model training, thus leaving more data for subsequent down-stream analyses. We acknowledge that the exact sample size requirements for domain adaptation may depend on a number of factors, including number of input features, multimodality, predictability of the brain embedding, and more. Thus, a more systematic investigation of different scenarios may be needed in a dedicated study. We have highlighted this in the discussion section of our manuscript (p.12, l. 319f.):

"However, we consider it important to further investigate the frameworks` boundaries in terms of sample characteristics of the source and target datasets, that is, under which condition the model performance diminishes."

(2) *The biological validation has different sample size requirements. In principle, since we are applying a machine learning model to individual MRI data, predictions are made at the single subject level and do not impose a sample size limitation. However, in subsequent association analyses there may be a requirement for larger samples to detect small effects. These requirements will depend on the studied phenotypes and the strength of the effects under investigation. In our case (puberty and mental health), the observed effect sizes ranged between small to moderate, which required sample sizes in the range of what is provided in ABCD and HBN for these effects to be detectable. For phenotypes where stronger effects are expected, the required sample sizes may well be smaller. The sample size requirement characteristics are, however, not unique to our work, but rather a general phenomenon in the field of psychiatric research, where we are dealing with a multitude of small effects (cf. Paulus and Thompson, 2019; DOI: 10.1001/jamapsychiatry.2018.4540; Gordon 2021; DOI: 10.1159/000517267).*

4. A table providing some information about the data used in the current study across datasets would be useful, this could be supplemental. Need a table for PNC, HBN and ABCD with information on %female and PDS scores/%F and M in different categories as well as information about no of diagnoses for M and F participants to better understand findings that were significant in F but not M in HBN dataset.

We have now added two tables called ‘Frequencies of pubertal categories in the ABCD sample’ and ‘Frequencies of psychopathology measures in the HBN sample’ to the Supplementary Material and refer to it in the main text:

“The frequency of pubertal categories for the baseline and follow-up data is shown in Supplementary Table 4.” (p.21, l. 600f.)

“Frequencies of psychopathology measures can be derived from Supplementary Table 5.” (p.21, l. 611f.)

Supplementary Table 4: Frequencies of pubertal categories in the ABCD sample.

Pubertal Category	Caregiver Report		Youth Report	
	male (N= 4045)	female (N= 3487)	male (N= 3761)	female (N= 2562)
baseline visit				
prepubertal	n= 2907 (71,9%)	n= 1129 (32,4%)	n= 1134 (30,2%)	n= 679 (26,5%)
early pubertal	n= 932 (23,0%)	n= 818 (23,5%)	n= 1783 (47,4%)	n= 697 (27,2%)
mid pubertal	n= 184 (4,5%)	n= 1441(41,3%)	n= 779 (20,7%)	n= 1097 (42,8%)

late pubertal	n= 20 (<1%)	n= 86 (2%)	n= 57 (1,5%)	n= 82 (3%)
postpubertal	n= 2 (<1%)	n= 4 (<1%)	n= 8 (<1%)	n= 7 (<1%)
2 years follow up visit				
	male (N=3970)	female (N= 3378)	male (N= 4095)	female (N= 3299)
prepubertal	n= 1448 (36,5%)	n= 147 (4,4%)	n= 878 (21,4%)	n= 162 (4,9%)
early pubertal	n= 1490 (37,5%)	n= 323 (9,5%)	n= 1715 (41,9%)	n= 360 (10,9%)
mid pubertal	n= 834 (21%)	n= 1601 (47,4%)	n= 1342 (32,8%)	n= 1563 (47,4%)
late pubertal	n= 194 (4,9%)	n= 1249 (37%)	n= 155 (3,8%)	n= 1166 (35,3%)
postpubertal	n= 4 (<1%)	n= 58 (1,7%)	n= 5 (<1%)	n= 48 (1,5%)

Supplementary Table 5: Frequencies of psychopathology measures in the HBN sample.

	male (N=1487)	female (N=784)
Number of diagnoses		
1	n= 384 (25,8%)	n= 202 (25,8%)
2	n= 420 (28,2%)	n= 231 (27,2%)
3	n= 300 (20,2%)	n= 163 (20,8%)
4	n= 175 (11,8%)	n= 98 (12,5%)
5	n= 107 (7,2%)	n= 58 (7,4%)
6	n= 56 (3,8%)	n= 31 (4,0%)
7	n= 31 (2,1%)	n= 16 (2,0%)
8	n= 7 (< 1%)	n= 3 (< 1%)
9	n= 3 (< 1%)	-
10	n= 4 (< 1%)	-
mean (SD)	2.71 (1.62)	2.71 (1.55)
Primary Diagnosis		
ADHD	n= 787 (52,9%)	n= 305 (38,9%)
ASD	n= 144 (9,7%)	n= 25 (3,2%)
Anxiety	n= 190 (12,8%)	n= 196 (25,0%)
Mood	n= 48 (3,2%)	n= 62 (7,9%)
Other Disorder	n= 68 (4,6%)	n= 33 (4,2%)
Other ND	n= 250 (16,8%)	n= 163 (20,8%)
CBCL Total		
mean (SD)	n= 1269 43,93 (25,99)	n= 635 42,60 (25,39)

Note: ADHD= Attention Deficit Hyperactivity Disorder, ASD= Autism Spectrum Disorder, ND= Neurodevelopmental Disorder; SD = standard deviation.

Furthermore, we have now also added a Supplementary Figure 6 referring to the HBN table below, which shows the distribution of the number of diagnoses per sex, both as total counts (Panel A) and as percentage values (Panel B).

Supplementary Fig. 6: Distribution of psychopathology measures in the HBN sample as total counts and percentages stratified for sex. Diagnoses were grouped into six distinct categories following Voldsbekk et al. (2023). ADHD= Attention Deficit Hyperactivity Disorder, ASD= Autism Spectrum Disorder, ND= Neurodevelopmental Disorder

Supplementary References:

- Voldsbekk, I. et al. *Delineating disorder-general and disorder-specific dimensions of psychopathology from functional brain networks in a developmental clinical sample.* <http://medrxiv.org/lookup/doi/10.1101/2023.03.31.23288009> (2023) doi:10.1101/2023.03.31.23288009.

Please note, that since we consider the PNC as a training sample for model building, we decided against adding an additional table dedicated at PNC since all relevant basic information like sample size, male-female ratio and age are already included in the Method section of our manuscript.

Reviewer #1 (Remarks to the Author):

The authors have addressed all of my concerns. Congrats to the authors on a great paper!
- Elvisha Dhamala

Reviewer #2 (Remarks to the Author):

I thank the authors for thoughtfully and comprehensive response to the suggestions. I feel this makes for a much stronger manuscript.

Reviewer #3 (Remarks to the Author):

I really appreciate the very thoughtful and comprehensive response to reviewer comments. No further concerns.